Resource

# A carnivorous plant genetic map: pitcher/insect-capture QTL on a genetic linkage map of *Sarracenia*

Russell L Malmberg[1,2] , Willie L Rogers[1], Magdy S Alabady[1,3]

The study of carnivorous plants can afford insight into their unique evolutionary adaptations and their interactions with prokaryotic and eukaryotic species. For *Sarracenia* (pitcher plants), we identified 64 quantitative trait loci (QTL) for insect-capture traits of the pitchers, providing the genetic basis for differences between the pitfall and lobster-trap strategies of insect capture. The linkage map developed here is based upon the F2 of a cross between *Sarracenia rosea* and *Sarracenia psittacina*; we mapped 437 single nucleotide polymorphism and simple sequence repeat markers. We measured pitcher traits which differ between *S. rosea* and *S. psittacina*, mapping 64 QTL for 17 pitcher traits; there are hot-spot locations where multiple QTL map near each other. There are epistatic interactions in many cases where there are multiple loci for a trait. The QTL map uncovered the genetic basis for the differences between pitfall- and lobster-traps, and the changes that occurred during the divergence of these species. The longevity and clonability of *Sarracenia* plants make the F2 mapping population a resource for mapping more traits and for phenotype-to-genotype studies.

## Introduction

Insect-eating plants fascinate scientists and the general public; Darwin was so taken with their unique adaptations that he used the word "wonderful" 15 times in his descriptions of them (Darwin, 1888). The carnivorous behavior of plants evolved independently, possibly as many as nine times (Givnish, 2015; Wheeler & Carstens, 2018). These organisms live in nutrient poor conditions, such as wetland bogs, and are able to obtain minerals, primarily nitrogen and phosphorous, from capturing and digesting insect prey (Darwin, 1888; Ellison & Adamec, 2018; Adlassnig et al, 2012; Bradshaw & Creelman, 1984). Their leaves are specialized to perform multiple functions; secrete attractive scents (Jurgens et al, 2009), capture insects, secrete extracellular digestive enzymes, absorb nutrients, photosynthesize, and develop microbial symbioses. Comparative genomic approaches

are beginning to be applied to identify sequences associated with the evolution of carnivory (Wheeler & Carstens, 2018). However, until this report, a classic genetic linkage mapping approach has not been implemented with these systems; we have used pitcher plants of the genus *Sarracenia* to develop such a genetic map.

*Sarracenia* pitcher plants are potentially useful for addressing a number of developmental, physiological, ecological, and evolutionary questions. Some of these are: How the pitchers actually digest the insects and obtain nutrition is of interest; for example, the flow of nitrogen and phosphorous (Gallie & Chang, 1997; Ellison, 2006; Butler & Ellison, 2007; Karagatzides et al, 2009, 2012) has been studied. Pitcher plants provide an opportunity to study their unique interactions with microbial communities; they contain a microbiome associated with insect digestion within their pitchers (Buckley et al, 2003; Gotelli & Ellison, 2006; Koopman et al, 2010; Koopman & Carstens, 2011; Baiser et al, 2012), a partially enclosed container offering an experimentally manipulable system suitable for studying eukaryotic-host/microbiome relationships. The plants and their pitchers can be used to study the developmental, evolutionary, and ecological tradeoffs associated with phenotypic plasticity as under high nitrogen conditions, the leaves will grow as flattened phyllodia (a leafstalk flattened into a blade shape which is photosynthetically more efficient than a pitcher [Ellison & Gotelli, 2002]) instead of as a pitcher. There are 14–50 named taxa within the genus (Schnell & McPherson, 2011; Weakley, 2015), and these can all hybridize with each other, yielding abundant material for investigating the role of hybridization in plant evolution. Some of these taxa are endangered, posing conservation genetics issues, such as how much genetic diversity is present in given populations, and what is the best way to preserve it? All of these example areas of inquiry would benefit from the availability of a genetic linkage map.

Pitcher plants are also charismatic attractive plants which can captivate broad audiences and help raise awareness of scientific issues in the general public. For an overview—a lively recent book covers many aspects of *Sarracenia* and carnivorous plant biology in detail (Ellison & Adamec, 2018).

Some of the difficulties in using *Sarracenia* as a genetic mapping system are: (a) it takes 3–4 y for a plant to produce seeds, (b) 1 cM on a genetic map would correspond on average to more than $10^6$

---

[1]Department of Plant Biology, Miller Plant Sciences Building, University of Georgia, Athens, GA, USA    [2]Institute of Bioinformatics, Davison Life Sciences Building, University of Georgia, Athens, GA, USA    [3]Georgia Genomics and Bioinformatics Core, University of Georgia, Athens, GA, USA

Correspondence: malmberg@uga.edu
Willie L Rogers' present address is State Botanical Garden of Georgia, University of Georgia, Athens, GA, USA

basepairs because of the large genome size (Rogers et al, 2010), and (c) the presence of a recent partial genome duplication (Srivastava et al, 2011) will complicate identifying orthologs. On the other hand: (a) the species/taxa can all be crossed with each other, and natural hybrids have been found for most of the combinations; (b) pitcher plants are long-lived perennials and can be readily vegetatively propagated, so that after genotyping individuals, a mapping population can be reused multiple times and the same genotype grown under multiple growth conditions for phenotyping.

We report developing an F2 generation of more than 200 individuals from a cross between *Sarracenia rosea* (*Sarracenia purpurea venosa burkii*) and *Sarracenia psittacina*, and use this F2 to create a genetic map with markers and pitcher quantitative trait loci (QTL). *S. rosea* was chosen as a parent as it is the Southeastern U.S. variant of *S. purpurea*, the most common, most familiar, and widely distributed of the *Sarracenias* in North America. *S. psittacina* was chosen as it is the most morphologically diverse and distinctive of the *Sarracenias*, with a decumbent, horizontal, pitcher growth habit; its pitchers have been described as lobster-trap–like with respect to catching insects, meaning that the insects can enter readily but have difficulty exiting. The other *Sarracenias* can be described as having pitfall traps into ponds. *S. rosea* and *S. psittacina* are in different subgroups of the genus, based upon a recently developed phylogeny of the taxa within the genus (Stephens et al, 2015) generated from the sequences of 199 nuclear genes. *S. rosea* groups with the other *S. purpurea*-related taxa, whereas *S. psittacina* groups with *Sarracenia minor*–related and *Sarracenia flava*–related taxa; hence, it is reasonable to expect some genetic differentiation between *S. rosea* and *S. psittacina* suitable for creating a genetic map. A third subgrouping within the genus contains *Sarracenia oreophila*, *Sarracenia rubra*–related taxa, *Sarracenia alata,* and *Sarracenia leucophylla.*

Some of the pitcher traits differing between *S. rosea* and *S. psittacina* have previously been investigated to identify their roles in insect attraction and capture (Horner et al, 2018). For example, the striking patterns of red stripes on pitchers, and other carnivorous plants, have been suggested to be involved in insect attraction (Schaefer & Ruxton, 2008). Many *Sarracenia* also have white windows or fenestrations in patterns on the pitchers which have been hypothesized to confuse trapped insects (Moran et al, 2012). There are morphological differences between *S. psittacina* pitchers and the pitchers of other *Sarracenia*, including the size and orientation of the orifice, the shapes of the hoods above and around the orifice, the presence of tissue and hairs apparently meant to trap the insects within the pitchers, and the overall small size of the *S. psittacina* pitcher (Naczi, 2018); Naczi made the intriguing observation and suggestion that *S. psittacina* pitchers are more similar to juvenile pitchers of other *Sarracenia*, and hence that their developmental evolution may be characterized as heterochrony. These pitcher differences between *S. rosea* and *S. psittacina* are thus some of the possible candidate traits for QTL genetic mapping.

We used the methods of genotyping by sequencing and restriction site–associated RNA-sequencing (RARseq [Alabady et al, 2015]) to generate linkage groups and a genetic map with more than 430 mapped markers covering 2017 cM. We identified and mapped 64 QTL for 17 pitcher traits which differ between the two parental species and which give insights into the genetic bases of the pitfall versus lobster-trap insect-capture strategies of *Sarracenia*.

# Results

## Linkage map

Hecht (1949) has described the chromosomes in root tip metaphases of six *Sarracenia* species, including *S. purpurea* and *S. psittacina*, as 2N = 26 or 1N = 13; a finished genetic map might thus have 13 linkage groups. Figs 1 and 2 show the genetic linkage map we developed, with linkage groups shown that are greater than 10 cM, or that are shorter than 10 cM but which also have a QTL mapping to them. Most of the markers we used for genetic mapping were single nucleotide polymorphisms (SNPs) identified by sequencing from RNA (RARseq) or DNA (RADseq), plus there were a few simple sequence repeat markers (SSRs). The final total of 437 is made up of 343 RNA-based SNPs, 78 DNA-based SNPs, and 16 SSRs. These are distributed into 42 linkage groups with a total length of 2,017 cM; 12 of the linkage groups contain a small number of markers and are less than 10 cM in length. The overall average density is 1 marker every 4.6 cM. The 13 largest linkage groups contain 273 markers with a total length of 1,446 cM, and with the shortest of the 13 having a length of 76.7 cM. The DNA-based markers were generally interspersed among the RNA-based markers, with 39 of them either a single isolated DNA marker or a doublet of two DNA markers. There were a few clusters of DNA-based markers: linkage group 24 was comprised entirely of 6 RADseq markers covering 20 cM whereas linkage group 27 was entirely 3 RADseq markers over 13 cM; the end of linkage group 1 contained 5 RADseq markers over 37 cM; near one end of linkage group 8 there was a group of 5 RADseq markers and 1 SSR over 28 cM.

The STACKS program (Catchen et al, 2013) generated sequence tags of 143 bases for each of the SNPs reported, as listed in Supplemental Information 5. We performed Blast2Go (Conesa et al, 2005; Gotz et al, 2008) analyses of these sequence tags for the SNPs (Supplemental Information 7). We could not detect a particular pattern in the functional annotations that would suggest anything about the nature of the sequences containing SNPs either overall or just the SNPs located near the mapped QTLs.

We did notice five instances of SNP tag sequence similarity where the SNPs were located in different regions of the genetic map, suggesting gene duplications. These included:

R7834 (lg3-pos32) and RI736 (lg3-pos59) with 100% identity over 143/143 bases;
R6009 (lg3-pos139) and R6000 (lg16-pos27) with 90% identity over 72/143 bases;
R8080 (lg4-pos40) and R8078 (lg4-pos70) with 100% identity over 120/143 bases;
R7818 (lg7-pos99) and R7820 (lg22-pos20) with 90% identity over 128/143 bases;
R1768 (lg8-pos0) and R1769 (lg15-pos0) with 100% identity over 97/143 bases.

## Pitcher traits mapped in F2

We measured a variety of pitcher traits that are different between *S. rosea* and *S. psittacina* to attempt to map their genetic bases. Fig 3 shows the parental plants and F1, giving an overall impression

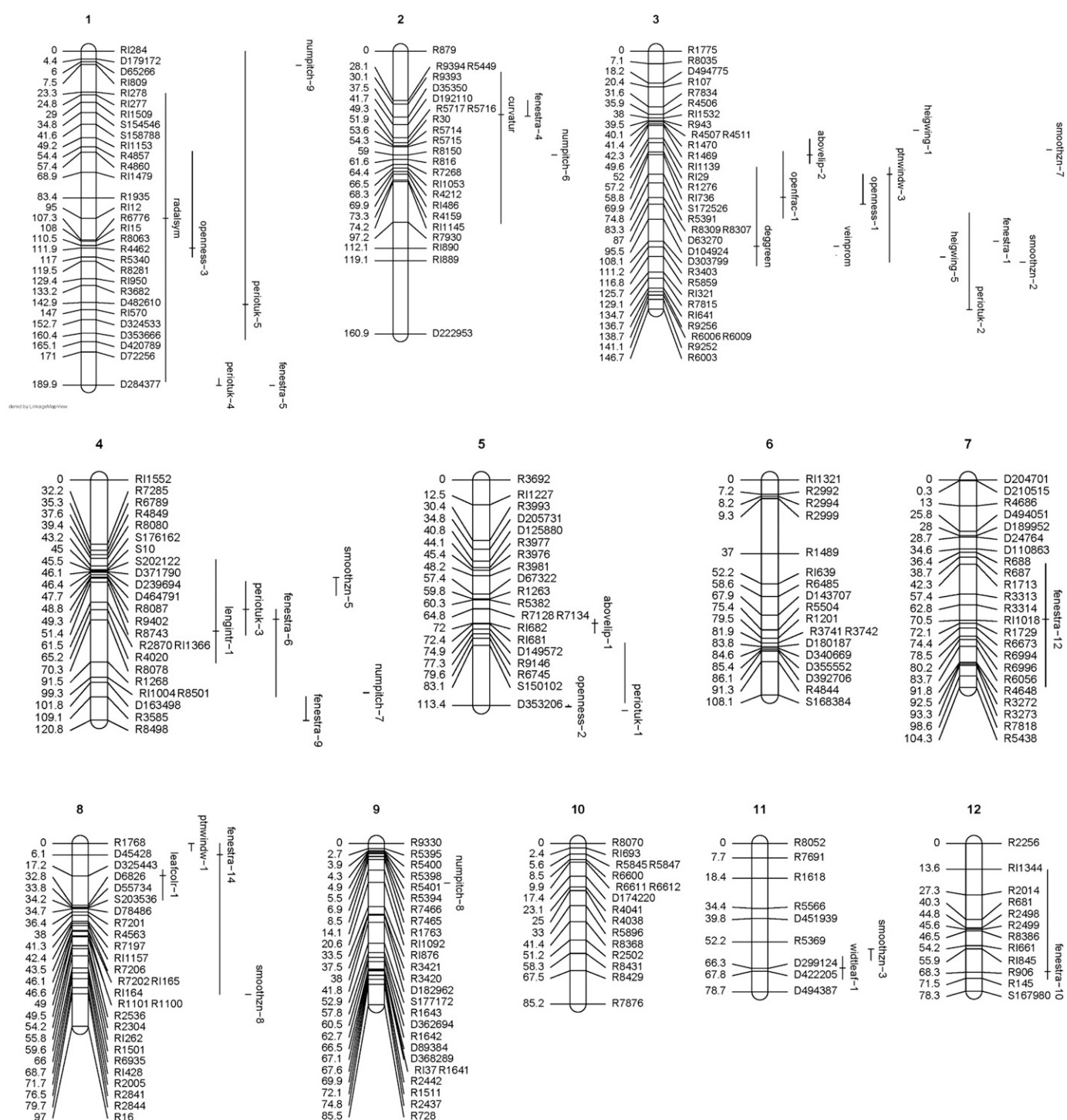

**Figure 1. Linkage groups with QTLs mapped (continued in Fig 2).**

Linkage groups are shown and numbered in decreasing order of length in cM. Linkage groups 1–30 are shown as they are greater than 10 cM in length; in addition groups 31, 34, and 39 are shown as QTLs mapped to them. QTLs are indicated by their name and by vertical lines which indicate a Bayesian position confidence interval, as calculated by R/qtl (Broman et al, 2003; Broman & Sen, 2009; Arends et al, 2010). When there are multiple loci for the same quantitative trait, these are indicated by a numbered trait name, which are numbered in decreasing order of %PVE such that number one indicates the locus with the highest %PVE. The genetic map was rendered by the R/LinkageMapView package (Ouellette et al, 2018).

Source data are available for this figure.

**Figure 2. Linkage groups with QTLs mapped (continued from Fig 1).**
Linkage groups are shown and numbered in decreasing order of length in cM. Linkage groups 1–30 are shown as they are greater than 10 cM in length; in addition groups 31, 34, and 39 are shown as QTLs mapped to them. QTLs are indicated by their name and by vertical lines which indicate a Bayesian position confidence interval, as calculated by R/qtl (Broman et al, 2003; Broman & Sen, 2009; Arends et al, 2010). When there are multiple loci for the same quantitative trait, these are indicated by a numbered trait name, which are numbered in decreasing order of %PVE such that number one indicates the locus with the highest %PVE. The genetic map was rendered by the R/LinkageMapView package (Ouellette et al, 2018).
Source data are available for this figure.

of the plants. For each quantitative trait mapped, the R/qtl fitqtl() function gives an estimate of the total phenotypic variance which is explained (%PVE, the heritability of the trait) in the F2 population by the postulated loci and the pairwise interactions, epistasis, between them. We describe here the traits mapped and the genetic loci and interactions involved, giving them in descending order of percent phenotypic variance explained; the eight letter codes shown are those used in the data files and in Table 1. When more than one locus is involved, the loci are numbered with locus-1 having the highest individual %PVE, and then descending in order; thus for traitx, traitx-1 has the highest individual %PVE, traitx-2 has the second highest, and so on. Figs 4, 5, 6, 7, 8, and 9 illustrate each of the individual pitcher traits measured. Table 2 summarizes the results for each trait and locus, whereas Table 3 lists the epistatic interactions detected.

There were nine traits measured which failed to show any heritability; these are described in the Materials and Methods section for their value as negative data.

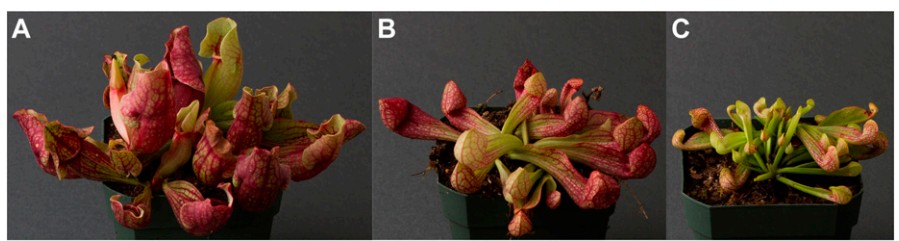

**Figure 3. Pitchers of the parents and the F1.**
**(A)** *S. purpurea.* **(B)** F1. **(C)** *S. psittacina.*
Source data are available for this figure.

**Table 1.  The 17 quantitative traits where a locus or loci mapped, listed in the order discussed in the text.**

| 8 letter code | In words |
|---|---|
| numpitch | Number of pitchers per rosette |
| smoothzn | Zone lacking hairs in pitcher |
| fenestra | White color windows on pitchers |
| periotuk | Periostome inward or outward |
| heigwing | Height of wing |
| ptnwindw | Pattern of color |
| openness | Degree of openness of pitchers |
| lengintr | Internal length of hairy region |
| veinprom | Well-defined veins |
| abovelip | Tissue above the lip line |
| widtleaf | Leaf width below opening |
| openfrac | Fraction of pitchers open |
| leafcolr | Leaf color ignoring veins |
| deggreen | Degree of green color |
| widtspot | Leaf width at widest spot |
| curvatur | Pitcher curvature |
| radalsym | Rosette radial symmetry |

Details of trait measurement are given in the results.

numpitch: Number of pitchers/leaves per rosette; 100% PVE; 9 loci plus 36 of 36 pairwise interactions (Fig 4A–C).

This is a count of the leaves/pitchers on a single rosette. *S. rosea* usually totals 4–6 leaves whereas *S. psittacina* can have up to 30. There are nine loci, and all pairwise interactions are significant. The pairwise interactions were individually small in their %PVE but totaled to 39% for the 36 pairs. The dominance relationships were: no dominance—loci 4, 6, 7, and 9; *S. rosea* dominance—locus 8; *S. psittacina* dominance—loci 1, 2, 3, and 5. The allele effect values were mixed across the multiple loci, with some in the direction of *S. rosea* and some in the direction of *S. psittacina*.

smoothzn: Smooth area lacking hairs (trichomes) within pitcher; 97% PVE; 9 loci plus 36 of 36 pairwise interactions (Fig 4D–G).

In *S. rosea* there is a glossy smooth zone with a polished look directly above the hairy zone at the base of the pitcher. *S. psittacina* lacks this smooth zone entirely; sometimes at the lip there can be a very small area of waxy surface but never this smooth zone. The F2s may have sporadic or plentiful hairs in a smooth zone, or pores for the hairs but no actual hairs seen. Nine significant loci were identified, all with similar %PVE. All the pairwise interactions were significant; however, the sizes of the %PVE for each of the interactions were small. For the 36 interactions, the sum of their %PVE was 32%. The dominance patterns were: no dominance—loci 2, 4, and 7; over/under-dominance—loci 1, 6, and 9; *S. rosea* dominance—locus 8; *S. psittacina* dominance—loci 3 and 5. The allele effect values were mixed across the multiple loci, with some in the directions of each parent, but more in the direction of *S. rosea*.

fenestra: White windows on the pitcher; 58%–99% PVE; 15 loci plus 65/105 interactions (Fig 4H–K).

*S. psittacina* pitchers have a windowing-effect pattern of white areas, whereas *S. rosea* pitchers do not. This was scored on a 6-point scale with 0 being *S. psittacina*–like, and 6 being *S. rosea*–like. A total of 13 significant loci were identified with 2 additional loci included as having significant interactions with other loci. The number of possible pairwise interactions among the 15 loci was too large to describe in a single genetic model to be evaluated by fitqtl (); the 55% PVE listed for the full model, and the values shown for the individual loci were obtained from a purely additive model. We explored the pairwise interactions by testing subsets of them in groups together with the additive effects; in some cases the final % PVE was as high as 99%; 65/105 interactions were significant. In the purely additive model, the locus *fenestra-1* explained 15% of the phenotypic variance, whereas other loci explained 7% or less. The dominance relationships were: no dominance—loci 1, 2, 3, 4, 7, 10, and 15; over/under-dominance—loci 8, 11, and 12; *S. rosea* dominance—loci 5 and 14; *S. psittacina* dominance—loci 6, 9, and 13. The allele effect values were mixed across the loci, with some in the directions of each parent.

periotuk: Periostome in or out; 50% PVE; 5 loci plus 1 of 10 pairwise interactions (Fig 5A–F).

Is the periostome tucked inward, *S. psittacina*–like as part of the lobster-trap morphology, or protruded, *S. rosea*–like as part of the pitfall morphology? This was scored on a 4 point scale with *S. psittacina*–like inward tucking as 0, *S. rosea*–like protrusion as 3, and with intermediates scored as 1 or 2. There were five significant loci, with one of the possible pairs, *periotuk-2* and *periotuk-3*, having a significant interaction. *Periotuk-1* and *periotuk-5* have dominance by the *S. psittacina* allele; *periotuk-2* and *periotuk-3* show no dominance; *periotuk-4* has dominance by the *S. rosea* allele. The allele effect values were generally in the direction of *S. rosea*.

heigwing: Height of wing; 43% PVE; five loci (Fig 5G–I).

This is the height of the wing perpendicular to the pitcher from its highest point directly back to where it meets and fuses with the leaf/body of the leaf. *Heigwing-1* and *heigwing-3* had no dominance; *heigwing-2* and *heigwing-5* had over- or underdominance; *heigwing-4* had dominance by the *S. rosea* allele. The allele effect values were mixed in the directions of both parents.

ptnwindw: Pattern of Color; 30% PVE; three loci plus two of three pairwise interactions (Fig 6A–C).

This is a measure of the patterning of the color with respect to veins and windows (fenestrations) on the pitcher. The color can be either green or a shade of purple. It was scored on a 0 to 3 point scale, where 0 is a solid color with veins not easily distinguishable (*S. rosea*), three indicates the veins and windows are highly noticeable (*S. psittacina*), and one and two are intermediate states. *Ptnwindow-1* and *ptnwindow-3* have dominant *S. psittacina* alleles, whereas *ptnwindow-2* shows no dominance. *Ptnwindow-1* has

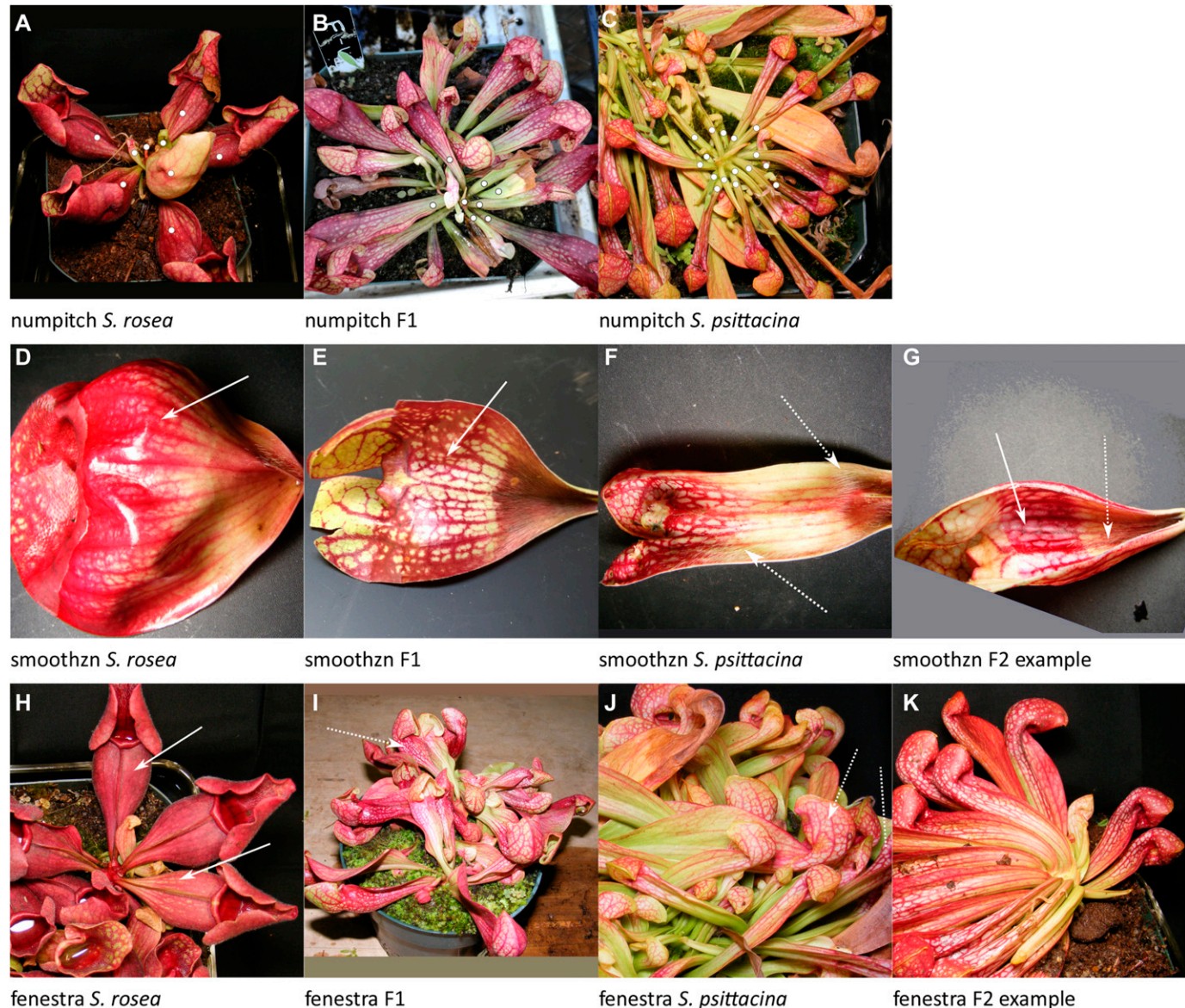

numpitch *S. rosea*　　numpitch F1　　numpitch *S. psittacina*

smoothzn *S. rosea*　　smoothzn F1　　smoothzn *S. psittacina*　　smoothzn F2 example

fenestra *S. rosea*　　fenestra F1　　fenestra *S. psittacina*　　fenestra F2 example

**Figure 4.　Examples of traits scored.**
Listed in order of decreasing %PVE, the same order as discussed in the text. **(A–C)** Dots indicate pitchers in rosette to be counted. **(D, E)** The solid arrow points to the glossy zone free of hairs. **(F)** The dashed arrow points to hairs throughout. **(G)** The solid arrow points to the smooth zone in pitcher otherwise more similar to *S. psittacina*. **(H)** The arrows point to absent or weak windowing effect. **(I, J)** The arrow points to white window effect. **(K)** Well-defined white windows due to distinct color borders. Source data are available for this figure.

significant interactions with each of *ptnwindow-2* and *ptwindow-3*. The allele effect values were in the direction of *S. psittacina*.

openness: Degree of openness of pitchers; 26% PVE; three loci (Fig 6D–F).

This measures the degree of openness of the operculum to the pitcher mouth, and is related to, but different, from *openfrac* which is the fraction of pitchers open. In *S. rosea* these are always open (pitfall trap), whereas in *S. psittacina* they are nearly always sealed closed except for a small opening (lobster-trap). The degree of openness was measured on a six point scale with five being

*S. rosea*–like and zero being *S. psittacina*–like. There were three significant loci. *Openness-1* and *openness-2* both have dominance by the *S. psittacina* allele, whereas *openness-3* shows no dominance. The allele effect values were in the directions of *S. psittacina*.

lengintr: Internal length hairy region; 23% PVE; two loci plus zero of one pairwise interaction (Fig 6G–I).

This is the distance from the base of the leaf where it attaches to the meristem upward until where the line of hairs ends, measured in pitchers which have been sliced open. In *S. rosea* this is at the very base of the leaf and does not extend very far upward; there is usually

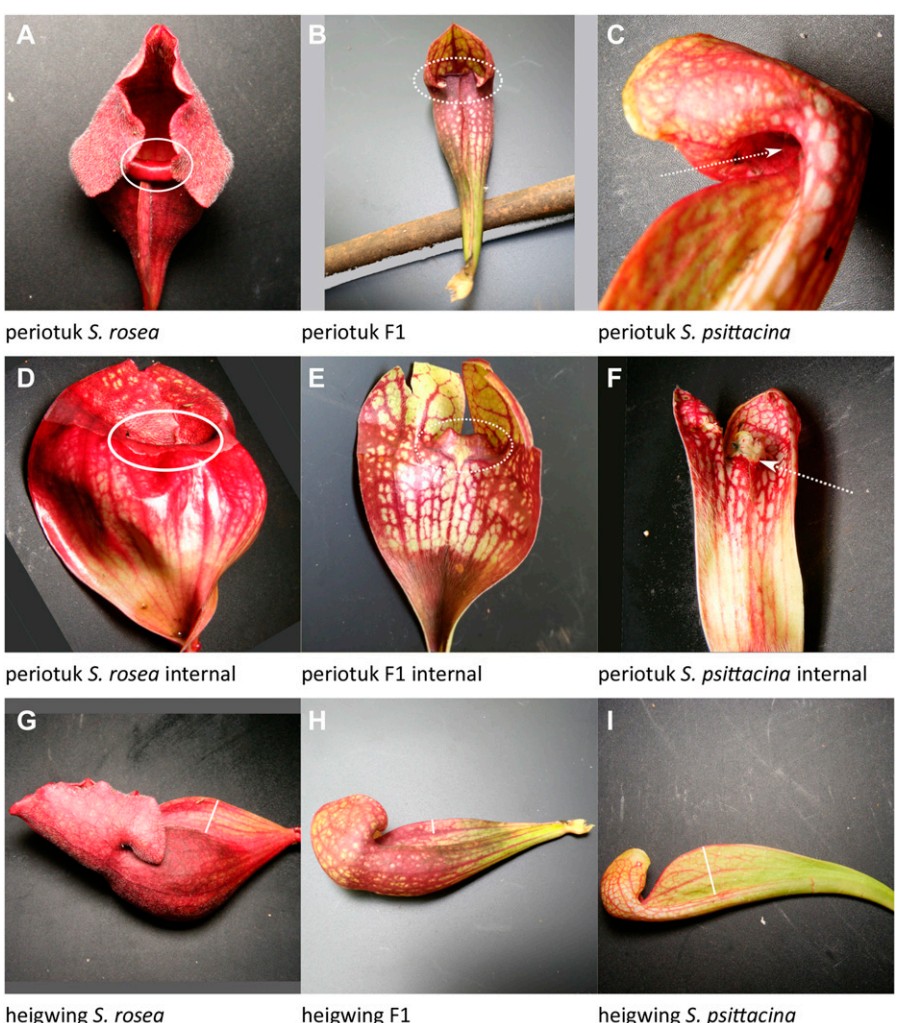

**Figure 5. Examples of traits scored.**
Listed in order of decreasing %PVE, the same order as discussed in the text. **(A)** The lip is curled outward. **(B)** The lip extends slightly upward. **(C)** The lip extends upward and inward. **(D)** There is no visible tissue above lip. **(E)** There is a small amount of tissue upward from lip. **(F)** There is tissue extending upward and inward from lip.
Source data are available for this figure.

periotuk *S. rosea*  periotuk F1  periotuk *S. psittacina*

periotuk *S. rosea* internal  periotuk F1 internal  periotuk *S. psittacina* internal

heigwing *S. rosea*  heigwing F1  heigwing *S. psittacina*

a pronounced clearly visible line at the base signaling the end of the hairy section. In *S. psittacina* this includes the entire length of the leaf right up to the peristome. The F2s have a wide range of region lengths and can have sporadic hairs appearing out of place. There are two loci, with the *lengintr-1* showing no dominance and *lengintr-2* showing partial dominance by the *S. psittacina* direction.

veinprom: Well-defined veins; 18% PVE; one locus (Fig 7A–C).

*S. psittacina* tends to have well-defined easily traceable veins, whereas *S. rosea* can have veins which are less clearly visible in the context of the leaf. This was scored on a 3 point scale with 0 being most well-defined, and 2 being least defined. There was one significant locus with no dominance.

Abovelip: Tissue above the lip line; 17% PVE; two loci plus one of one pairwise interaction (Fig 7D–F).

This measures the length of the tissue above the lip line. When one slices the leaves open (down the back of the leaf and filet the leaf open), *S. psittacina* has tissue extending upward protruding into the bell that makes the lobster-trap. When one does the same

thing to *S. rosea* there is no tissue extending upward but instead tissue curled down and under to form the lip rim as part of the pitfall trap. There are two significant loci with a significant interaction between them. Considered individually, *abovelip-1* has dominance by the *S. psittacina* allele, whereas *abovelip-2* has dominance by the *S. purpurea* allele; the highest value occurs in individuals heterozygous for both loci.

widtleaf: Leaf width, 17% PVE, two loci plus one of one pairwise interaction (Fig 7G–I).

This is the width of the leaf directly below the mouth opening. There are two significant loci, and the interaction between the two is also significant. The locus effects were in opposing directions with the largest widths in pitchers homozygous for the *widtleaf-1* locus from *S. psittacina* and *widtleaf-2* locus from *S. rosea*. There is no significant dominance.

openfrac: Fraction of pitchers open (pitfall) versus mostly closed (lobster); 16% PVE; two loci plus zero of one pairwise interaction (Fig 8A–D).

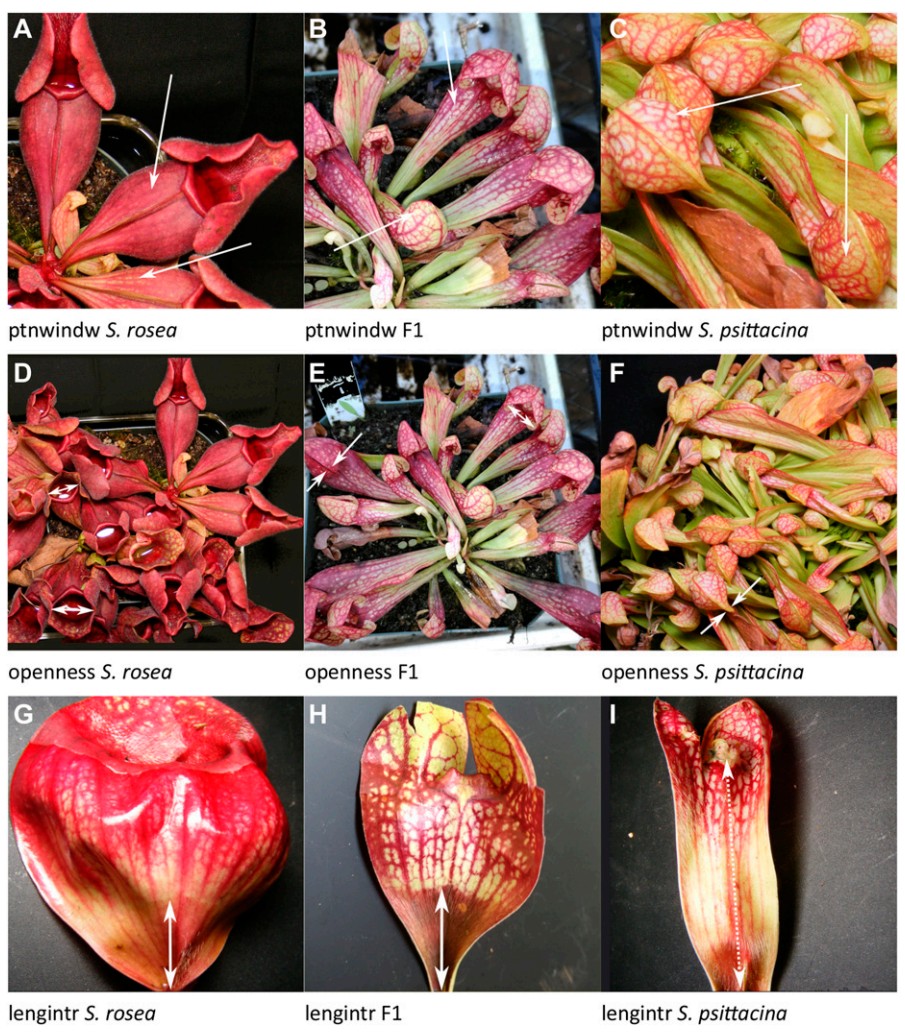

**Figure 6. Examples of traits scored.**
Listed in order of decreasing %PVE, the same order as discussed in the text. **(A–C)** Do the veins and windows have similar or different color patterns? **(D–F)** The degree of openness of pitcher. **(G–I)** The length of the region containing hairs from base upward.
Source data are available for this figure.

This is the fraction of pitchers which are open, *S. rosea* pitfall-like, versus mostly closed with just a small periostome opening at the base, *S. psittacina* or lobster-trap–like. There are two loci, with no interactions; *openfrac-2* shows dominance by the *S. psittacina* allele.

leafcolr: Leaf color, green versus purple, ignoring veins; 16% PVE; two loci plus one of one pairwise interaction (Fig 8E–G).

This is the degree of purpleness, with green scored as 0, intense purple scored as 3, and intermediate shades as 1 or 2. Two significant genetic loci were found, with an interaction between them. The most intense color was found when the two loci were both homozygous for the *S. psittacina* allele, but the least intense was when *leafcolr-1* was homozygous for the *S. psittacina* allele and *leafcolr-2* was homozygous for the *S. rosea* allele. *Leafcolr-1* had some dominance by *S. psittacina*, and *leafcolr-2* showed no dominance.

deggreen: Degree of green color; 8% PVE; one locus (Fig 8H–K).

This is an overall estimate of how much green versus red there is in an individual. In some F2 plants, individual pitchers could vary significantly from each other. Plants were scored on a 0 to 3 point scale where 0 is all

green and 3 is no green. There was one locus found, explaining 8% of the phenotypic variation, with dominance by the *S. rosea* allele.

widtspot: Widest leaf width; 8% PVE; one locus (Fig 9A–C).

This is the width of the leaf at its widest spot, which may be either above or below the mouth, whereas the widtleaf measure is always just below the mouth opening, and maps to different loci. The one locus found shows dominance by the *S. psittacina* allele.

curvatur: Pitcher curvature, 6% PVE; one locus (Fig 9D–F).

This was measured as the angle between a flat area on the back of the pitcher and the tip of the operculum. For *S. rosea*, this number is usually 90° but for *S. psittacina* it is always less than 50°. One significant locus was found which explained 6% of the phenotypic variation. There is no evidence for dominance.

radalsym: Rosette radial symmetry, 4% PVE, one locus (Fig 9G–J).

This is a visual observation of whether or not the rosettes have radial symmetrical looking downward at them. In some non-symmetric

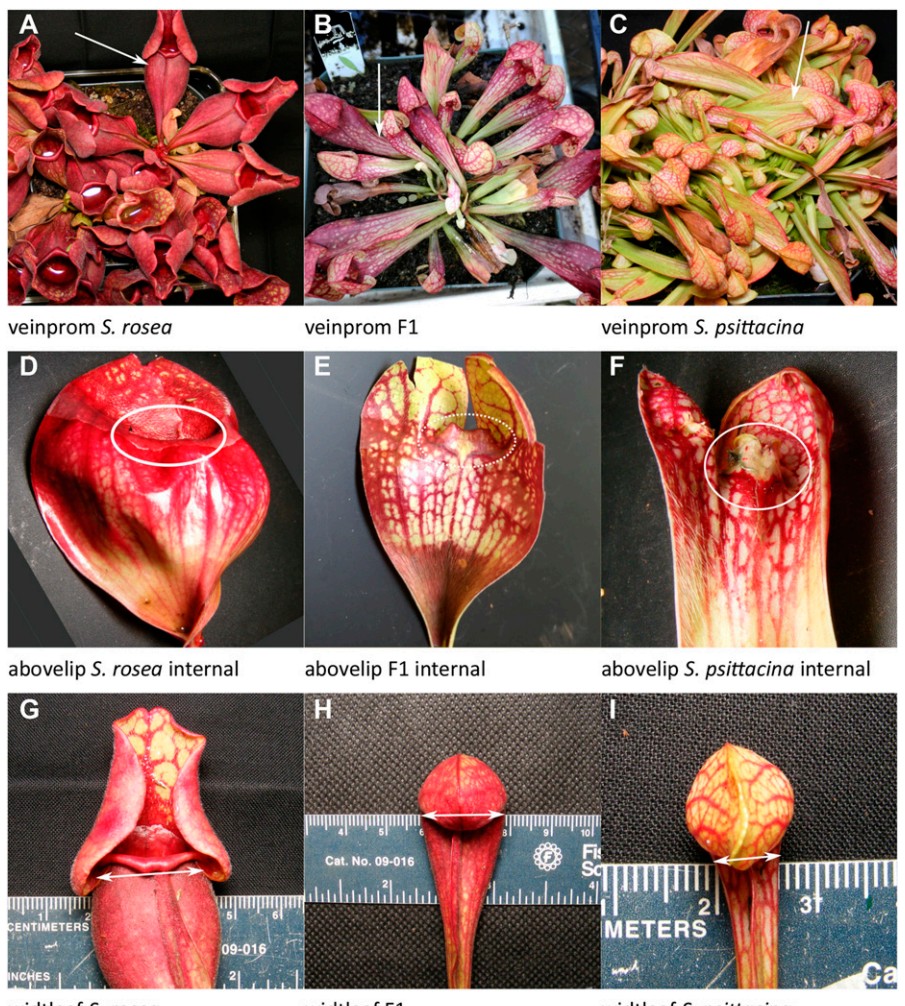

**Figure 7. Examples of traits scored.**
Listed in order of decreasing %PVE, the same order as discussed in the text. **(A–C)** How well defined and easily traceable are the veins? **(D–F)** The length of tissue above the lip line. **(G–I)** The width of the leaf directly below the mouth opening.
Source data are available for this figure.

veinprom *S. rosea*    veinprom F1    veinprom *S. psittacina*

abovelip *S. rosea* internal    abovelip F1 internal    abovelip *S. psittacina* internal

widtleaf *S. rosea*    widtleaf F1    widtleaf *S. psittacina*

F2 plants, the pitchers bunch together more on one side of the rosette than the other; or the pitchers may project upward and outward at irregular angles. This was scored as a binary trait, 1 or 0. A single significant genetic locus was found which explained 4% of the phenotypic variation. *S. rosea* has greater symmetry than *S. psittacina*; and this trait had significant dominance by the *S. psittacina* direction.

### Traits in the F1

The F1 plants were largely intermediate in all traits examined between the two parental species (Fig 3). Many of the traits we measured had multiple loci detected, with a mix of dominance relationships and allele effects; hence, the intermediate state of the F1 is not surprising.

# Discussion

We have constructed a linkage map of *Sarracenia* using an F2 generation of a cross between two species, *S. psittacina* and a

subspecies of *S. purpurea*, *S. rosea*. For 17 pitcher traits which differ between these taxa, we identified 64 loci which were placed on the genetic map. This is the first genetic linkage map and QTL mapping for a carnivorous plant.

### A genetic linkage map for *Sarracenia*

Our linkage map contains 437 markers across 42 linkage groups, with a total length of 2,017 cM. Twelve of these linkage groups were 10 cM or less in length. Previous cytogenetic work (Hecht, 1949) indicated that there are 1N = 13 chromosomes. The 13 longest of the linkage groups we have identified range from 189.9 cM to 76.6 cM with a total length of 1,446 cM. For comparison, grape, *Vitis*, is a species with extensive sequencing, and its transcript sequences are some of the most similar to those of *Sarracenia* transcripts as detected by Blast comparisons (Srivastava et al, 2011). A recent *Vitis* linkage map (Wang et al, 2017) had 6,000 markers with a total length of 2,186 cM in 19 linkage groups with lengths ranging from 86 cM to 237 cM. A linkage map of maize (Davis et al, 1999), certainly one of the genetically most well-studied plants, contains 10 linkage groups representing 10 chromosomes, and has a total length of 1,723 cM;

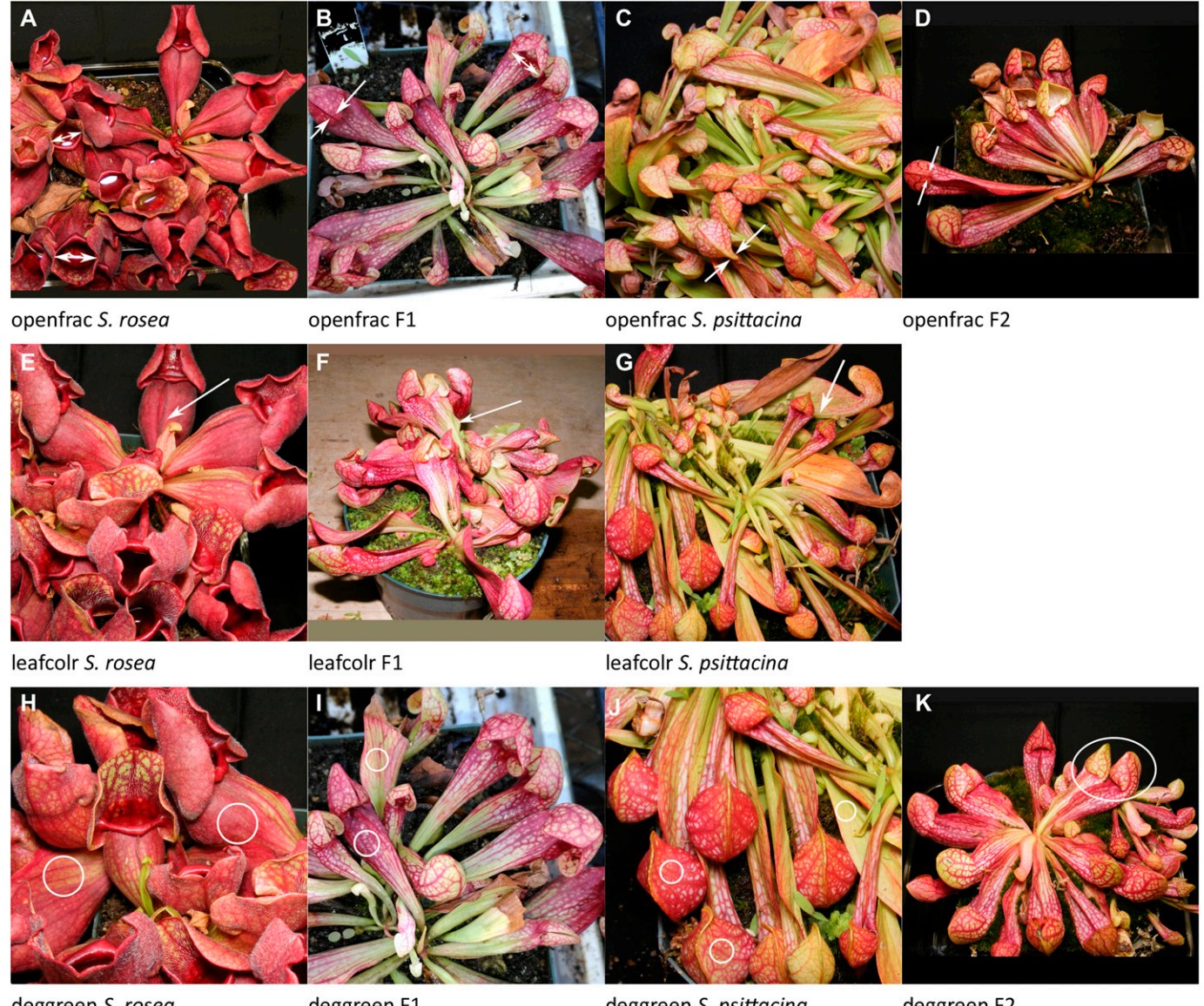

**Figure 8. Examples of traits scored.**
Listed in order of decreasing %PVE, the same order as discussed in the text. **(A–D)** The fraction of pitchers open. **(E–G)** Green versus red/purple leaf color. **(H–K)** The fraction of pitchers with green versus red/purple leaves.
Source data are available for this figure.

the maize genome is about 70% of the size of the *Sarracenia* genome. These comparisons suggest that our current linkage map covers most of the total genetic map, more than 80% of it, and lead to a rough global estimate of 1.4 × 10⁶ bp per cM for *Sarracenia*.

There were several difficulties in constructing the genetic map: it took a long time, more than 10 y, to generate the F2 plants. The parents were heterozygous at multiple loci as we did not start with pure lines as parents (Mendel, 1951). *Sarracenia* has a relatively large genome size estimated to be 3.6 × 10⁹ basepairs (Rogers et al, 2010). There is a recent partial genome duplication which dates to about the time of the divergence of the species within the genus (Srivastava et al, 2011), perhaps 2 million years ago. These difficulties likely lead to the map being incomplete in spite of our multiple rounds of

sequencing efforts. Although we were successful in generating some SNP markers which were able to be mapped with the RADseq protocol (Elshire et al, 2011) (78 SNPs), the RARseq (Alabady et al, 2015) method of generating markers from RNA was more productive (343 SNPs). A few of the RNA-based markers we mapped showed evidence of duplications. The existence of high quality reference genome sequences from both the two parents, *S. rosea* and *S. psittacina* would likely greatly improve the genetic map.

## Pitcher trait inheritance

The focus of our trait measurements has been on pitcher traits as insectivory is a hallmark of the genus, and insectivorous

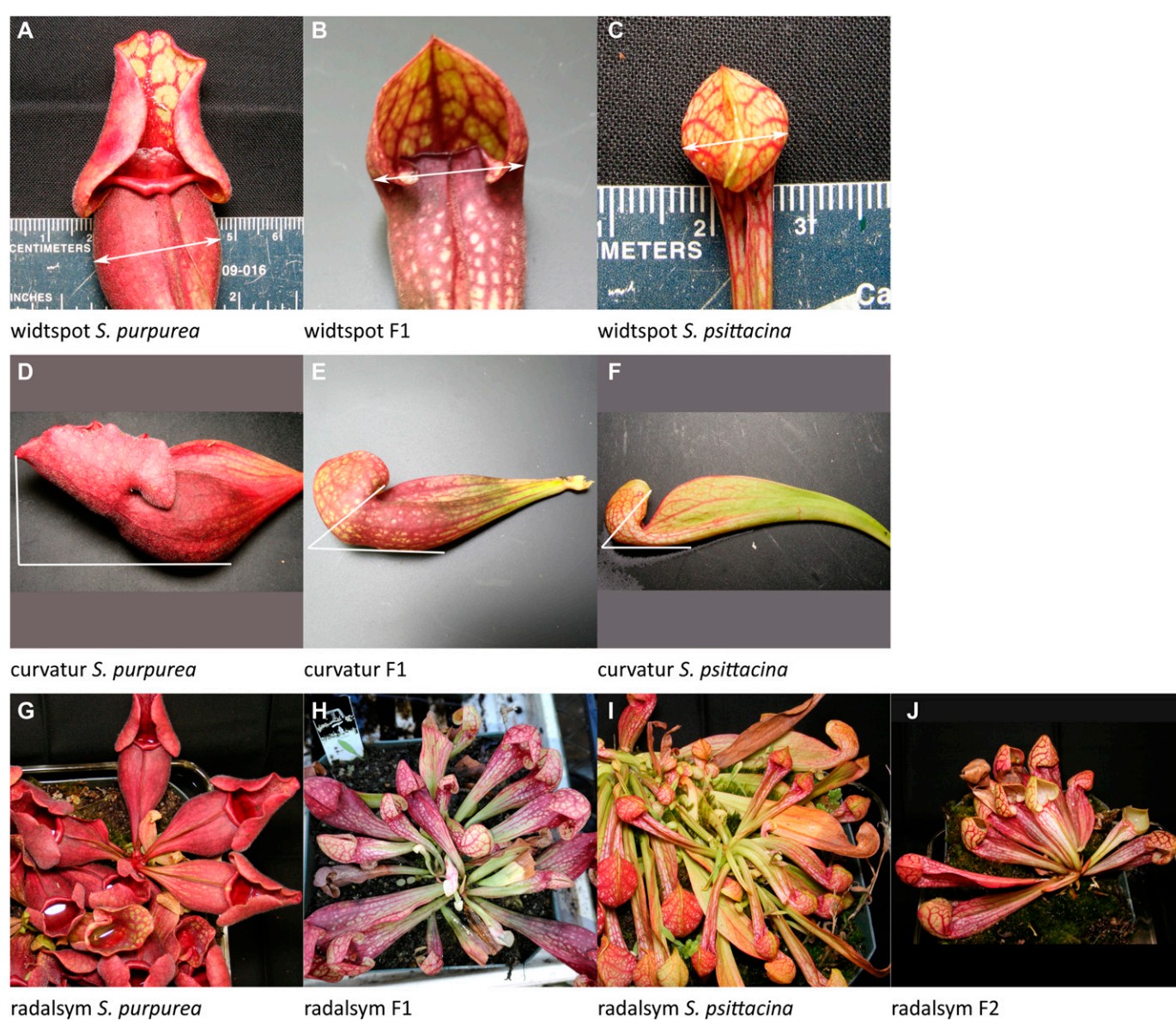

**Figure 9. Examples of traits scored.**
Listed in order of decreasing %PVE, the same order as discussed in the text. **(A–C)** The width of the leaf at its widest spot. **(D–F)** The angle between a flat area on the back of the pitcher and the tip of the operculum. **(G–J)** The degree of radial symmetry of rosette, with asymmetry seen in F2.
Source data are available for this figure.

adaptations are the reason for the general fascination with these plants. There are a wide diversity of pitcher morphologies within the *Sarracenia* genus, which may be associated with their insect-eating lifestyle, and may represent differences in insect-catching and digestion strategies across the species. We picked 26 different pitcher traits to measure; for 17 of these traits we were able to map significant QTLs, whereas nine of them failed to map. Our current study established the genetic system and was an initial exploration of quantitative trait heritability in one particular setting. Any of these traits may behave differently when the plants are grown in a different environment. Some of the traits which showed no in-heritance or low %PVE could have higher heritability if studied

somewhat differently; conversely a different environment might reduce the %PVE for other traits.

The %PVE varied from 3.7% (radalsym) to 96.5% (smoothzn) and 99.6% (numpitch) for the full models. Radalsym was presence or absence of a lack of radial symmetry in the pitcher rosettes (Fig 9G–J); both parents have greater radial symmetry than some of the F2 plants which showed the most extreme phenotype. Smoothzn is the glossy internal region free of hairs found in *S. rosea* (Fig 4D–G), whereas numpitch is the number of leaves/pitchers per rosette (Fig 3A–C). The next highest %PVE after numpitch and smoothzn was 55.3% (periotuk) (Fig 5A–F). The high %PVE may partially be due to these two traits being straightforward to score in

**Table 2. Quantitative traits mapped and their loci.**

| QT model | QT loci | Linkage group | Nearest marker | Position max. | Position interval | LOD | % PVE | P-value | Significance | Effect | Effect SD | Dominance |
|---|---|---|---|---|---|---|---|---|---|---|---|---|
| numpitch-full | | | | | | 246.2 | 99.6 | 0.000 | *** | | | |
| | numpitch-1 | 26 | D163468 | 3.8 | 3.8–3.8 | 165.1 | 16.1 | 0.000 | *** | 0.03 | 1.01 | T |
| | numpitch-2 | 30 | S151516 | 10.4 | 10.4–10.4 | 163.4 | 15.4 | 0.000 | *** | 1.21 | 0.99 | T |
| | numpitch-3 | 15 | R1047 | 14.9 | 14.9–14.9 | 163.4 | 15.4 | 0.000 | *** | −0.62 | 1.02 | T |
| | numpitch-4 | 18 | RI18 | 22.2 | 22.2–22.2 | 163.1 | 15.3 | 0.000 | *** | 0.09 | 1.03 | — |
| | numpitch-5 | 31 | RI787 | 0.0 | 0.0–0.0 | 161.0 | 14.6 | 0.000 | *** | 0.67 | 1.29 | T |
| | numpitch-6 | 2 | R8150 | 59.0 | 59.0–59.0 | 154.3 | 12.5 | 0.000 | *** | −0.60 | 1.34 | — |
| | numpitch-7 | 4 | R3585 | 106.8 | 106.8–106.8 | 153.3 | 12.3 | 0.000 | *** | 0.57 | 0.99 | — |
| | numpitch-8 | 9 | RI1092 | 20.6 | 20.6–20.6 | 142.2 | 9.5 | 0.000 | *** | −0.42 | 0.96 | R |
| | numpitch-9 | 1 | RI809 | 7.5 | 7.5–7.5 | 136.3 | 8.3 | 0.000 | *** | 0.70 | 1.13 | — |
| smoothzn-full | | | | | | 163.9 | 96.5 | 0.000 | *** | | | |
| | smoothzn-1 | 40 | R8758 | 0.0 | 0.0–0.0 | 73.1 | 12.1 | 0.000 | *** | 2.39 | 1.13 | O |
| | smoothzn-2 | 3 | R5859, RI321 | 120.0 | 120.0–120.0 | 71.3 | 11.5 | 0.000 | *** | −0.24 | 2.32 | — |
| | smoothzn-3 | 11 | R5369 | 56.0 | 56.0–62.0 | 70.0 | 11.1 | 0.000 | *** | 4.46 | 1.52 | T |
| | smoothzn-4 | 22 | S177374 | 2.0 | 2.0–2.0 | 69.6 | 11.0 | 0.000 | *** | −0.06 | 2.25 | — |
| | smoothzn-5 | 4 | R9402 | 49.3 | 48.8–58.0 | 67.8 | 11.5 | 0.000 | *** | 3.27 | 1.43 | T |
| | smoothzn-6 | 16 | R4706 | 32.0 | 32.0–32.0 | 67.5 | 11.4 | 0.000 | *** | 1.28 | 1.97 | O |
| | smoothzn-7 | 3 | RI29, R1276 | 56.0 | 56.0–56.0 | 67.5 | 11.4 | 0.000 | *** | 0.17 | 1.19 | — |
| | smoothzn-8 | 8 | R2844 | 79.7 | 79.7–79.7 | 66.0 | 10.0 | 0.000 | *** | 2.92 | 1.45 | R |
| | smoothzn-9 | 29 | R3218 | 0.0 | 0.0–0.0 | 62.4 | 9.0 | 0.000 | *** | 0.38 | 2.32 | O |
| fenestra-full | | | | | | 39.6 | 55.3 | 0.000 | *** | | | |
| | fenestra-1 | 3 | D303799 | 108.1 | 108.1–108.1 | 14.8 | 15.7 | 0.000 | *** | −0.84 | 0.15 | — |
| | fenestra-2 | 26 | D163468 | 3.8 | 3.8–8.2 | 7.5 | 7.4 | 0.000 | *** | −0.53 | 0.14 | — |
| | fenestra-3 | 15 | R1048 | 12.5 | 12.5–12.5 | 7.4 | 7.3 | 0.000 | *** | −0.40 | 0.15 | — |
| | fenestra-4 | 2 | D35350 | 37.5 | 28.1–37.5 | 5.6 | 5.4 | 0.000 | *** | −0.30 | 0.14 | — |
| | fenestra-5 | 1 | D284377 | 189.9 | 189.9–189.9 | 5.4 | 5.2 | 0.000 | *** | −0.39 | 0.13 | R |
| | fenestra-6 | 4 | R8078 | 70.3 | 65.2–109.1 | 4.7 | 4.5 | 0.000 | *** | 0.26 | 0.17 | T |
| | fenestra-7 | 34 | R600 | 1.0 | 1.0–6.0 | 4.1 | 3.9 | 0.000 | *** | −0.22 | 0.13 | — |
| | fenestra-8 | 18 | R6464 | 47.3 | 43.6–47.3 | 3.6 | 3.4 | 0.001 | *** | −0.02 | 0.15 | O |
| | fenestra-9 | 4 | R8498 | 120.8 | 109.1–120.8 | 3.4 | 3.2 | 0.001 | ** | 0.14 | 0.26 | T |
| | fenestra-10 | 12 | R906 | 68.3 | 13.6–71.5 | 3.2 | 3.0 | 0.002 | ** | 0.17 | 0.16 | — |
| | fenestra-11 | 13 | R7097 | 59.9 | 0.0–59.9 | 2.6 | 2.5 | 0.005 | ** | 0.19 | 0.13 | O |
| | fenestra-12 | 7 | RI1018 | 70.0 | 42.3–104.3 | 2.6 | 2.4 | 0.006 | ** | −0.07 | 0.12 | O |
| | fenestra-13 | 19 | R4633 | 20.9 | 20.9–38.6 | 2.3 | 2.1 | 0.011 | * | −0.20 | 0.13 | T |
| | fenestra-14 | 8 | D45428 | 6.1 | 0.0–79.7 | 1.5 | 1.4 | 0.053 | | −0.29 | 0.14 | R |
| | fenestra-15 | 17 | R2675 | 27.5 | 0.0–54.8 | 0.8 | 0.7 | 0.215 | | −0.24 | 0.12 | — |

| QT model | QT loci | Linkage group | Nearest marker | Position max. | Position interval | LOD | % PVE | P-value | Significance | Effect | Effect SD | Dominance |
|---|---|---|---|---|---|---|---|---|---|---|---|---|
| periotuk-full | | | | | | 34.4 | 50.3 | 0.000 | *** | | | |
| | periotuk-1 | 5 | S150112, D353206 | 116.0 | 82.0–112.0 | 12.3 | 14.1 | 0.000 | *** | 0.29 | 0.12 | T |
| | periotuk-2 | 3 | R6003 | 146.7 | 92.0–146.7 | 11.9 | 13.7 | 0.001 | *** | 0.34 | 0.10 | — |
| | periotuk-3 | 4 | R4020 | 65.2 | 51.4–78.0 | 11.6 | 13.2 | 0.001 | *** | 0.19 | 0.10 | — |
| | periotuk-4 | 1 | D284377 | 189.9 | 186.0–189.9 | 11.6 | 12.0 | 0.002 | ** | 0.01 | 0.12 | R |
| | periotuk-5 | 1 | D482611, RI570 | 144.0 | 0.0–164.0 | 9.3 | 11.3 | 0.012 | * | 0.27 | 0.13 | T |
| heigwing-full | | | | | | 27.9 | 43.4 | 0.000 | *** | | | |
| | heigwing-1 | 3 | R1469 | 45.2 | 45.2–45.2 | 12.7 | 16.7 | 0.000 | *** | 2.41 | 0.64 | — |
| | heigwing-2 | 30 | S151516 | 10.4 | 10.4–10.4 | 11.2 | 14.5 | 0.001 | ** | −0.96 | 0.52 | O |
| | heigwing-3 | 16 | R1856 | 41.2 | 13.3–58.6 | 10.1 | 12.9 | 0.005 | ** | 1.79 | 0.47 | — |
| | heigwing-4 | 17 | D197566 | 18.0 | 18.0–18.0 | 9.2 | 11.7 | 0.011 | * | −0.79 | 0.56 | R |
| | heigwing-5 | 3 | R5859 | 116.8 | 116.8–116.8 | 8.3 | 10.5 | 0.028 | * | −0.65 | 0.67 | O |
| ptnwindw-full | | | | | | 17.6 | 30.2 | 0.000 | *** | | | |
| | ptnwindw-1 | 8 | R1768 | 0.0 | 0.0–4.0 | 12.6 | 20.4 | 0.000 | *** | −0.08 | 0.08 | T |
| | ptnwindw-2 | 34 | R600, D285722 | 2.0 | 0.6–6.0 | 11.5 | 16.7 | 0.000 | *** | −0.23 | 0.11 | — |
| | ptnwindw-3 | 3 | R5391 | 70.0 | 66.0–120.0 | 8.5 | 13.3 | 0.000 | *** | −0.12 | 0.07 | T |
| openness-full | | | | | | 14.7 | 25.8 | 0.000 | *** | | | |
| | openness-1 | 3 | D63270 | 87.0 | 69.9–87.0 | 10.5 | 17.7 | 0.000 | *** | −0.56 | 0.13 | T |
| | openness-2 | 5 | D353206 | 114.1 | 113.4–114.1 | 5.6 | 9.0 | 0.007 | ** | −0.37 | 0.14 | T |
| | openness-3 | 1 | R8063 | 111.9 | 57.4–117.0 | 5.6 | 9.0 | 0.007 | ** | −0.33 | 0.14 | — |
| lengintr-full | | | | | | 12.9 | 23.2 | 0.000 | *** | | | |
| | lengintr-1 | 4 | R8078 | 76.0 | 40.0–92.0 | 9.1 | 15.7 | 0.000 | *** | −5.71 | 1.07 | — |
| | lengintr-2 | 14 | R934 | 37.4 | 29.1–56.0 | 5.7 | 9.4 | 0.000 | *** | −4.39 | 1.02 | T |
| veinprom | veinprom | 3 | R3403 | 111.2 | 116.0–116.0 | 9.9 | 18.3 | 0.000 | *** | 0.37 | 0.05 | — |
| abovelip-full | | | | | | 9.0 | 16.9 | 0.000 | *** | | | |
| | abovelip-1 | 5 | RI682 | 72.0 | 70.0–77.3 | 7.7 | 14.2 | 0.000 | *** | −0.34 | 0.19 | R |
| | abovelip-2 | 3 | RI736 | 58.8 | 49.6–64.0 | 7.2 | 13.2 | 0.000 | *** | 0.30 | 0.19 | T |
| widtleaf-full | | | | | | 9.1 | 16.9 | 0.000 | *** | | | |
| | widtleaf-1 | 11 | D299124 | 66.0 | 60.0–72.0 | 8.3 | 15.3 | 0.000 | *** | 2.20 | 0.52 | — |
| | widtleaf-2 | 22 | S177374 | 0.0 | 0.0–4.0 | 6.9 | 12.5 | 0.000 | *** | −0.39 | 0.52 | — |
| openfrac-full | | | | | | 7.9 | 16.2 | 0.000 | *** | | | |
| | openfrac-1 | 3 | R8309 | 83.3 | 57.2–95.5 | 6.0 | 11.9 | 0.000 | *** | 0.12 | 0.03 | — |
| | openfrac-2 | 20 | R6925 | 26.8 | 1.7–31.5 | 4.3 | 8.5 | 0.004 | ** | 0.09 | 0.04 | T |

**Table 2. Continued**

| QT model | QT loci | Linkage group | Nearest marker | Position max. | Position interval | LOD | % PVE | P-value | Significance | Effect | Effect SD | Dominance |
|---|---|---|---|---|---|---|---|---|---|---|---|---|
| leafcolr-full | | | | | | 8.6 | 16.0 | 0.000 | *** | | | |
| | leafcolr-1 | 8 | D325443 | 17.2 | 14.0–30.0 | 8.3 | 15.3 | 0.000 | *** | 0.21 | 0.12 | T |
| | leafcolr-2 | 18 | R1315 | 34.0 | 6.0–42.0 | 7.5 | 13.8 | 0.000 | *** | −0.14 | 0.13 | — |
| deggreen | deggreen | 3 | R3403 | 111.2 | 66.0–122.0 | 4.2 | 8.3 | 0.000 | *** | 0.29 | 0.07 | R |
| widtspot | widtspot | 18 | D126613 | 11.9 | 0.0–26.4 | 4.2 | 8.3 | 0.000 | *** | −0.32 | 0.08 | T |
| curvatur | curvatur | 2 | D35350 | 36.0 | 12.0–98.0 | 3.1 | 6.2 | 0.001 | *** | 2.67 | 0.74 | — |
| radalsym | radalsym | 1 | RI12 | 95.0 | 24.0–188.0 | 1.8 | 3.7 | 0.015 | * | −0.10 | 0.06 | T |

QTL Model: For traits where there are multiple loci identified, the full model is indicated on the first line of the group with a -full suffix, then with individual loci numbered below that. Traits are listed in decreasing order of percentage phenotypic variance explained; within a multi-locus trait, loci are numbered in decreasing order of percentage phenotypic variance explained. Trait codes are given in Table 1. Position Max refers to the most likely single position and Position Interval gives the range from a Bayesian confidence interval. LOD is the log-odds score, %PVE is the percentage phenotypic variance explained by that model or locus. In the case of the -full models, except for fenestra, this includes additive effects of each locus and all pairwise interactions. For individual loci for a trait which has multiple loci, the %PVE is taken from the R/qtl fitqtl() dropone analysis. For fenestra, there were too many possible pairwise interactions hence the full model and the individual loci are taken from a purely additive model; fenestra-14 and fenestra-15 are included because subsequent calculations of pairwise interactions indicated they had some significant interactions as listed in Table 3. P-value is the probability based upon an F-statistic; significance levels are indicated based upon * for <0.05, ** for <0.01, and *** for <0.001. Estimated additive allele effect and SD of the estimate. Dominance was determined by inspection of the R/qtl effectplot() function results (Supplemental Information 6): dash indicates semi-dominance; O indicates over- or under-dominance; T indicates *S. psittacina* allele dominance; R indicates *S. rosea* allele dominance.

a quantitative fashion. Other traits with lower %PVE may show more inheritance if scored differently or if the plants were grown differently. For single loci, the %PVE fell between 2 and 21%. The highest value was 20.4% for *ptnwindw-1*, one locus for a trait (Fig 6A–C) which measured the pattern of color in veins and non-vein areas. Some of the traits with multiple loci identified had a single locus which explained most of the variation together with additional loci of smaller effect; other traits had several loci with similarly sized %PVE. There was not a discernible pattern to the dominance relationships: semidominance, over- or under-dominance, and dominance from both parents were found. For the allele effects, when multiple loci were found for some traits, the pattern of effects sometimes pointed to one parent, but sometimes was mixed with direction pointing to both of the two parents.

Table 3 summarizes pairwise interactions among the loci where these were significant and the total %PVE explained by the interaction was greater than 1%. For the traits smoothzn, numpitch, and fenestra, which had large numbers of loci identified, the %PVE for any one interaction was small, on the order of 1%–2%.

The trait fenestra (Fig 4H–K), white windows on the pitchers, had the most loci (15) in the initial analysis. Because of computational limitations, we were not able to test all possible pairwise interactions among these in the same way as we could for the traits with smaller numbers of loci. We therefore tested a number of QTL models which included the additive and random subsets of the interaction possibilities. The values shown in Table 2 for fenestra are from evaluation of an additive model, not allowing any interactions. The bottom of Table 3 lists interactions among the fenestra loci which were potentially significant in the exploratory analyses.

The genetic map, Figs 1 and 2, shows the traits mapped with a Bayesian confidence window, as calculated by R/qtl (Broman et al, 2003; Arends et al, 2010). The position confidence intervals vary

from being very large, poorly defined and taking up most of a linkage group, to appearing very small and highly defined. There is an inverse relationship between the %PVE for whole trait, and the smallness of the confidence interval. Thus, *radalsym*, with the lowest %PVE, has its position poorly defined on a linkage group. In contrast, the numpitch and smoothzn loci appear to almost have point estimates of their position. The same F2 plants were scored for all the traits. A trait with a low %PVE effectively has less of its variation available to indicate the genetic map position than a trait with a high %PVE and hence has a more poorly defined position.

There are some intriguing overlapping localizations of loci from multiple traits appearing in the same map region. Linkage group 3 has 12 loci from smoothzn, fenestra, periotuk, heigwing, ptnwindow, openness, veinprom, abovelip, openfrac, and deggreen mapping in the same general vicinity. Several other such QTL hotspots seem to exist. Assuming that these hotspots are meaningful and not by chance, a variety of mechanisms might potentially be involved in creating and maintaining them: pleiotropy, the influence of a single gene on seemingly unrelated phenotypic traits (Plate, 1910; Stearns, 2010) can create multiple phenotypes from a small number of genes; selective sweeps, when a selectively favored allele increases the frequencies of nearby genetic variants by hitchhiking (Smith & Haigh, 1974), can lead to co-segregation of the linked genes, associating the phenotypes; chromosomal rearrangements, such as inversions, can suppress recombination within a region, also causing association of phenotypes.

### Pitfall and lobster traps

The genetic cross we performed for mapping was between *S. psittacina* and *S. rosea*. *S. rosea* is the southeastern variant of *S. purpurea*, and is a member of the medium-height group of

**Table 3.**  Interactions between loci for the same trait with >1% phenotypic variance explained.

| QT model | QTL pair | LOD | %PVE | P-value | Significance |
|---|---|---|---|---|---|
| leafcolr | 1:2 | 7.2 | 13.08 | 0.000 | *** |
| widtleaf | 1:2 | 6.0 | 11.85 | 0.000 | *** |
| abovelip | 1:2 | 5.5 | 9.91 | 0.000 | *** |
| ptnwindw | 1:3 | 5.9 | 8.87 | 0.000 | *** |
| ptnwindw | 1:2 | 5.9 | 8.85 | 0.000 | *** |
| smoothzn | 6:1 | 17.1 | 1.46 | 0.000 | ** |
| smoothzn | 2:6 | 15.5 | 1.31 | 0.001 | ** |
| smoothzn | 2:1 | 14.3 | 1.19 | 0.001 | * |
| smoothzn | 2:8 | 13.9 | 1.15 | 0.001 | * |
| smoothzn | 7:8 | 13.6 | 1.12 | 0.002 | * |
| smoothzn | 2:5 | 13.2 | 1.08 | 0.002 | * |
| smoothzn | 3:4 | 12.9 | 1.06 | 0.002 | * |
| smoothzn | 7:9 | 12.8 | 1.04 | 0.002 | * |
| smoothzn | 8:3 | 12.7 | 1.03 | 0.003 | * |
| smoothzn | 7:2 | 12.4 | 1.01 | 0.003 | * |
| smoothzn | 7:6 | 12.4 | 1.01 | 0.003 | * |
| numpitch | 8:5 | 87.9 | 2.54 | 0.000 | *** |
| numpitch | 9:2 | 87.5 | 2.51 | 0.000 | *** |
| numpitch | 6:1 | 84.2 | 2.30 | 0.000 | *** |
| numpitch | 9:3 | 76.8 | 1.89 | 0.000 | *** |
| numpitch | 7:4 | 73.5 | 1.73 | 0.000 | *** |
| numpitch | 1:2 | 72.9 | 1.70 | 0.000 | *** |
| numpitch | 7:2 | 69.8 | 1.56 | 0.000 | *** |
| numpitch | 2:5 | 68.2 | 1.49 | 0.000 | *** |
| numpitch | 6:5 | 67.9 | 1.48 | 0.000 | *** |
| numpitch | 4:5 | 67.3 | 1.45 | 0.000 | *** |
| numpitch | 7:1 | 65.2 | 1.36 | 0.000 | *** |
| numpitch | 8:3 | 64.9 | 1.35 | 0.000 | *** |
| numpitch | 3:4 | 62.7 | 1.27 | 0.000 | *** |
| numpitch | 7:5 | 57.9 | 1.10 | 0.000 | *** |
| numpitch | 6:3 | 55.9 | 1.03 | 0.000 | *** |
| numpitch | 9:5 | 55.5 | 1.02 | 0.000 | *** |
| numpitch | 1:5 | 54.9 | 1.00 | 0.000 | *** |

Special case—fenestra: The 65 possible interactions discovered among loci in decreasing order of significance were: 12:15, 12:4, 12:9, 14:3, 12:6, 11:4, 5:6, 12:11, 5:11, 7:4, 10:4, 7:15, 12:10, 3:6, 7:13, 10:3, 10:8, 1:9, 5:15, 14:7, 11:10, 7:10, 12:3, 7:8, 14:8, 5:8, 14:15, 5:9, 3:13, 1:7, 10:15, 1:15, 1:3, 14:11, 11:3, 3:4, 1:11, 7:6, 5:3, 10:6, 7:3, 7:11, 3:8, 1:6, 11:9, 1:2, 7:9, 2:5, 1:10, 12:8, 3:9, 14:10, 1:4, 13:9, 5:7, 7:12, 5:14, 1:13, 2:8, 14:6, 6:9, 2:3, 3:15, 2:11, 10:9.
QTL Model: See Table 1 for an explanation of the names. The QTL-pair refers to the locus numbers which are interacting. Interactions are listed which are significant and for which %PVE >1%, except for fenestra. LOD is the log-odds score. %PVE is the percentage phenotypic variance explained by that interaction. P-value is the probability; significance levels are indicated based upon * for <0.05, ** for <0.01, and *** for <0.001, where significance levels have been corrected (reduced) for multiple trait testing.

Sarracenia taxa. _S. psittacina_ pitchers have been described as lobster-trap–like with respect to catching insects, whereas the other _Sarracenia_ are described as having pitfall traps.

Some of the loci we have identified are part of the genetic basis for the lobster-trap versus pitfall morphologies. The periotuk trait describes the inward folded tissue of the lobster-trap, and we identified five loci involved in this, whereas abovelip measures the amount of tissue in this folding, and we found one locus for this. These are both lobster-trap traits from the _S. psittacina_ parent. The two open traits openness and openfrac describe how wide the pitcher opening is, with the wider pitcher being an _S. rosea_ pitfall trait, and between which we found a total of four loci. The inside surfaces of pitchers of _S. purpurea_

and *S. rosea* have a smooth, glossy, hair-free zone above a more hirsute region, whereas *S. psittacina* has hairs (trichomes) throughout. The presence of hairs is possibly important for *S. psittacina* to help keep the insects trapped inside, in combination with the narrow opening and inward periostome, whereas *S. rosea* has its pitcher pond, and thus may not need hairs in that region. Fig 4G is an example of an F2 pitcher which is largely *S. psittacina*–like in shape, but which has some smoothzn. Fenestration, the pattern of white windows, has been interpreted as being involved in insect prey-attraction (Schaefer & Ruxton, 2014). This is found in *S. psittacina*, not *S. rosea*; we found the fenestra quantitative trait has 13–15 loci and potentially a large number of interactions among them. The genetic loci underpinning the periotuk, openness, openfrac, smoothzn, and fenestra traits are thus involved in the pitfall versus lobster-trap differences in insect-capture strategy.

### Evolution

The phylogeny of the genus from Stephens et al (2015) divides the taxa into two main groups—tall pitcher plants (*S. oreophila*, *Sarracenia alabamensis*, *S. alata*, *S. rosea*, and related taxa) and mixed-sized pitcher plants (*S. purpurea*, *S. minor*, *S. psittacina*, *S. flava* and other taxa); *S. psittacina* is the shortest, smallest, and most horizontal in its growth habit. The radiation of the taxa within the genus was relatively recent, possibly 2 million years ago at a time co-incident with a partial genome duplication (Srivastava et al, 2011). Our analysis detected 64 trait loci which differ between *S. rosea* and *S. psittacina*. The analysis of crosses between similar plant species has shown examples of both a small number of major effect loci, and also a larger number of minor effect loci for traits contributing to the evolution or domestication of these species (as examples [Doebley, John, 1992; Wills & Burke, 2007; Brandvain et al, 2014; Fishman et al, 2014; Wang et al, 2015; Garner et al, 2016; Kenney & Sweigart, 2016; Badouin et al, 2017]). The pattern we have seen in the pitcher traits examined so far is more similar to having a large number of minor effect loci. We have not examined floral traits whose inheritance might be expected to provide insight into the divergence of *S. rosea* and *S. psittacina*; superficially, the flowers of the two parental species differ modestly in size and color, but their overall morphology is very similar. Particularly where there are multiple loci for the same trait, some of the genetic differences we have found might be related to duplications of portions of the genome, beyond the five pairs of loci listed which map at different locations. The multiple loci may be identifying paralogous genes.

### Prospects

We have generated a linkage map which is 80% complete; the successful mapping of a number of pitcher QTLs demonstrates some of the potential of this system. We have studied a variety of pitcher morphological traits, but there are many other questions that could be similarly addressed with these F2 plants by measuring appropriate traits, provided they differ between *S. rosea* and *S. psittacina*. As examples: The roles of the various pitcher components in insect capture and digestion might be studied in F2 plants which contain various combinations of the traits, such as hairs (trichomes), fenestrations, and periostome structures, to determine the effect of each; specific combinations of QTL could be used to test

results from the presence and absence of particular pitcher features. Our pitcher QTL were measured at only a relatively full grown stage of the pitchers; comparison of pitchers of *S. rosea*, *S. psittacina*, and the F2 plants at various developmental stages might help examine the hypothesis (Naczi, 2018) that heterochrony is a part of the *S. psittacina* developmental evolution. Hybridization can occur between all species or taxa within the genus, raising questions about what is the nature of a species in *Sarracenia*, and what evolutionary genetic mechanisms exist which might maintain a species in the face of rampant hybridization. The genetic map and the F2 population could be tools which contribute to solving these problems.

One of the attractions of this system is that the F2 plants are long-lived and are easily subdivided for vegetative propagation. Given a set of F2 plants which have been genotyped, it should be possible to readily map multiple additional QTLs by scoring the trait of interest across these plants, then performing a QTL analysis with the same marker and map files. For any traits of interest, these could be scored under multiple environmental conditions by growing clones of the characterized genotypes. Similarly, additional analyses might improve the number of markers in each F2 plant, and then it would be straightforward to reanalyze the data with a more detailed genetic map.

## Materials and Methods

### Cross

The initial cross was between the southeastern variety of *S. purpurea*, *S. rosea* (*S. purpurea venosa burkii*) (male), and *S. psittacina* (female). F1 plants were grown and selfed to generate F2 individuals. All plants were grown in the Plant Biology Department greenhouses, University of Georgia, Athens, Georgia, USA (33°55′45.7″N 83°21′49.6″W).

### RNA-based markers

We used the restriction site associated RARseq method, as described by Alabady et al (2015), to generate RNA-based SNP markers. RNA was isolated from each of the F2 plants then these were reverse transcribed into cDNA using the methods described (Alabady et al, 2015). The cDNA samples were digested with restriction *MseI*, then the resulting fragments were size fractionated on an agarose gel to produce fragments of 250–600 bp. These fragments were ligated to oligonucleotide adapters which contained unique barcodes for each individual plant and which prepared the samples as libraries for Illumina sequencing. The sequencing was performed on the NextSeq platform.

We generated reference transcriptome sequences from *S. rosea* and *S. psittacina* with deep coverage, using both nextseq (Illumina) and long read Iso-Seq (PacBio) platforms. SNP identification was performed using the program STACKS (Catchen et al, 2013) version 1.48; we performed reference-based SNP detection by comparison of the sequences from the F2 plants to the reference transcriptomes. In the final marker set used for mapping, the SNPs detected from the NextSeq ranscriptome reference begin with the letter R followed by the number assigned by STACKS. The actual

SNPs in their sequence contexts are given in the Supplemental Information 5.

### DNA-based markers

We used standard restriction site–associated DNA sequencing (RADseq) methods (Elshire et al, 2011) to generate a subset of genomic fragments for sequencing. A double digest with *MspI* and *SbfI* succeeded in generating sequences with a reasonable probability of being matched across the F2 individuals. After the size selection, the fragments were ligated to oligonucleotide adapters which contained unique barcodes for each individual plant and which prepared the samples as libraries for Illumina sequencing. The sequencing was carried out on the NextSeq platform. SNPs were detected by the software STACKS as a denovo, non-referenced–based, pipeline. In the final marker set used for mapping, the SNPs detected from RADseq begin with the letter D followed by the number assigned by STACKS. The actual SNPs in their sequence contexts are given in the Supplemental Information 5.

A total of 16 SSRs (microsatellites), isolated as described in Rogers et al (2010) were also used as markers in the map construction. The SSR primer sequences are given in the Supplemental Information 6.

### Traits

The F2 plants were greenhouse grown. The plant traits discussed were measured in May 2017, when most of the F2 plants were 5 y old. The traits are illustrated in Fig 4 (parents) and in Fig 5 (F2 examples).

In addition to the traits presented in the results and discussion, there were nine traits which failed to show any genetic component to the phenotypic variance: plant height (plheight); a combination of pitcher width and wing height measured in a profile (winglfpf); the distance from the back of the leaf to the tip of the operculum measured in a profile (pfloperc); the width of the pitcher at its widest point above the mouth opening, periostome, (widtbell); a measure of the location of green in a leaf of mixed color (locgreen); a measure of the color of the veins (veincolr); a measure of the thickness of the periostome lip (periowid); total length from base to crown (lengbscr); and one measure of whether the periostome was more *S. rosea* or *S. psittacina*—like (perioinv).

### Map construction

R version 3.4.3 2017-11-30 (R Core Team, 2013), and the packages that are associated with this version were used in this analysis. The R package onemap (Margarido et al, 2007) was used to construct the map, with some additional analyses of the linkage map performed by the R package qtl (Broman et al, 2003; Broman & Sen, 2009; Arends et al, 2010). Markers were discarded if they appeared in fewer than 10% of the total genetic lines; F2 genotypes were discarded if they had fewer than 10% of the total number of markers; any marker showing segregation distortion was discarded. The final dataset contained 281 F2 lines (genotypes), and 572 markers. The onemap function group() formed linkage groups, then order_seq() and ripple_seq() were used to form the maps. The onemap package

suggested an initial LOD score criterion of 6.1 for the mapping process; we actually used LOD 6.5.

Blast2Go (Conesa et al, 2005; Gotz et al, 2008) analysis of the sequence tags for the mapped SNP markers was performed using version 5.1.12 with a standard annotation pipeline of blast and interpro searches followed by go mapping and annotation. The summary results table (Supplemental Information 7) is organized in genetic map order with the marker and QTL positions indicated on the genetic map.

### QTL mapping

The R package qtl (Broman et al, 2003; Broman & Sen, 2009; Arends et al, 2010) was used for mapping the quantitative traits. The qtl package functions scanone() and scantwo() were used to scan for candidate loci, and then makeqtl() and fitqtl() were used to evaluate the qtl models. 2,000 permutations were used to set significance levels. Significant QTLs were identified for 17 of 26 quantitative traits. The ANOVA approach of the fitqtl() function was used to determine the significance of single or multiple loci per trait, and to report the significance of all pairwise interactions among these loci, or pairwise epistasis. Effectplot() was also used to help assess allele effects and dominance; examples are given in Fig S1. The QTL effects were calculated by R/qtl as 0.5 * (*rosea*_value - *psittacina*_value).

We tested some of the mqm multiple qtl mapping functions; however, the mqmaugment() function never finished its analysis, which suggests that this approach does not work well with our dataset (Broman in Google Groups R/qtl discussion, 2017). The traits and their scoring are discussed in more detail in the Results section. The package LinkageMapView (Ouellette et al, 2018) works together with the qtl package, and was used to generate the genetic map and QTL diagrams.

## Data Availability

The sequencing data from this publication have been deposited to the NCBI SRA database (https://www.ncbi.nlm.nih.gov/bioproject/) and assigned the identifier PRJNA288354 with SRA identifiers SRR8096761 and SRR8096762. The data required for genetic marker and QTL mapping are contained in Supplemental Information 2, 3, 4, 5, 6, 7 accompanying this manuscript.

## Supplementary Information

## Acknowledgements

The project was completed over a number of years and we had help and useful discussions with a number of individuals, including: Jess Stephens, Na Wang, Ed McAssey, Jim Leebens-Mack, Jim Hamrick, Greg Cousins, Mary Jo Godt, Lisa Donovan, Ron Determann, Kelly Dawe, Jenny Cruse-Sanders, Michael Boyd, and John Burke. Two manuscript reviewers provided helpful

comments and suggestions. Funding supporting this research was received from a University of Georgia Distinguished University Professorship research fund to RL Malmberg.

## Author Contributions

RL Malmberg: conceptualization, resources, data curation, software, formal analysis, supervision, funding acquisition, validation, investigation, visualization, methodology, project administration, and writing—original draft, review, and editing.
WL Rogers: data curation, investigation, methodology, project administration, and writing—review and editing.
MS Alabady: resources, data curation, software, formal analysis, investigation, methodology, and writing—review and editing.

## Conflict of Interest Statement

The authors declare that they have no conflicts of interest.

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
