## [Reviewer comments · Life Science Alliance]

Life Science Alliance

A Carnivorous Plant Genetic Map: Pitcher/Insect-Capture QTL On a Genetic Linkage Map of *Sarracenia*

Russell Malmberg, Willie Rogers, and Magdy Alabady
DOI: [10.26508/lsa.201800146](https://doi.org/10.26508/lsa.201800146)

Corresponding author(s): Russell Malmberg, University of Georgia

Review Timeline:	Submission Date:	2018-08-03
	Editorial Decision:	2018-09-24
	Revision Received:	2018-10-24
	Editorial Decision:	2018-11-13
	Revision Received:	2018-11-16
	Accepted:	2018-11-19

Scientific Editor: Andrea Leibfried

Transaction Report:

September 24, 2018

Re: Life Science Alliance manuscript #LSA-2018-00146-T

Dr. Russell L Malmberg
University of Georgia
Plant Biology Department
2502 Miller Plant Sciences Bldg
Athens, Georgia 30602-7271

Dear Dr. Malmberg,

Thank you for submitting your manuscript entitled "The First Carnivorous Plant Genetic Map: Pitcher/Insect-Capture QTL On A Genetic Map of Sarracenia" to Life Science Alliance. Please excuse the delay in getting back to you, I had to give the reviewers more time to assess your work because of the recent vacation period. The comments of two expert reviewers are appended to this letter.

As you will, the reviewers appreciate your data and provide constructive input on how to further strengthen your manuscript by introducing text changes and re-analysing data already at hand. We would thus like to invite you to provide a revised version, addressing all issues noted by the reviewers. Importantly, the directionality of QTL should be reported and the rationale/justification for this work and the evolutionary interest needs to be better described.

Thank you for this interesting contribution to Life Science Alliance. We are looking forward to receiving your revised manuscript.

Sincerely,

- A letter addressing the reviewers' comments point by point.
- An editable version of the final text (.DOC or .DOCX) is needed for copyediting (no PDFs).
- High-resolution figure, supplementary figure and video files uploaded as individual files: See our detailed guidelines for preparing your production-ready images, <http://life-science-alliance.org/authorguide>
- Summary blurb (enter in submission system): A short text summarizing in a single sentence the study (max. 200 characters including spaces). This text is used in conjunction with the titles of papers, hence should be informative and complementary to the title and running title. It should describe the context and significance of the findings for a general readership; it should be written in the present tense and refer to the work in the third person. Author names should not be mentioned.

B. MANUSCRIPT ORGANIZATION AND FORMATTING:

Full guidelines are available on our Instructions for Authors page, <http://life-science-alliance.org/authorguide>

Reviewer #1 (Comments to the Authors (Required)):

Malmberg et al. take a genetic mapping approach to determine the genetic architecture of 17 traits in a cross between two species of *Sarracenia* carnivorous plants that differ in pitcher morphotypes.

They discovered a total of 64 QTL distributed across the genome, several of which co-localize to similar linkage group locations, and uncovered several interactions between multiple loci affecting a single trait. The main advancement of this study seems to be the construction and application of the first genetic map for carnivorous plant species - a task the authors highlight as difficult given the life cycle of these species. However, this resource will be of use to future research given the ease of vegetatively propagating these species. The methods and analyses of this paper are robust, and the construction of the first genetic map for carnivorous plants is a significant contribution. Given these qualities, this paper should be of interest to those studying these species, and their ecological interactions. However, in my opinion, the manuscript could use improvement in major areas; framing and organization. In addition, I have added some minor comments. These are detailed below.

FRAMING

The framing of this study is not well developed in the introduction. In my opinion, the authors are correct that "insect-eating plants fascinate scientists and the general public" (line 29). They are inherently interesting, especially from an evolutionary and ecological perspective. However, just stating this, to me, does not seem to be a strong enough reasoning for why someone should be interested in this work. I would suggest more strongly introducing the interesting ecology behind carnivorous adaptations in plants early in the introduction, as a means to attract your readership.

The authors should discuss how these plants can thrive in low N soils, and expand on the phenotypic plasticity observed under varying levels of N (this is cool science, and would certainly add more interest). Likewise, they could expand on the tissue specialization brought up in lines 32-36.

Furthermore, consider adding a few sentences describing why the two focal species were specifically chosen: Is there some evolutionary divergence story that may be of interest? Are these representative "model" species for carnivorous plants? In other words, what makes these two species of particular interest.

ORGANIZATION & STRUCTURE

The next major item I wanted to touch on was overall organization and structure across the manuscript. First, I do not think that Table 1 is necessary for the main text. The authors present your QTL, their locations and intervals in Figure 1. Furthermore, they present the %PVE and interactions in the results text that describes each mapped trait. Table 1 seems mostly redundant, and can be tucked into the supplemental material.

Secondly, it is difficult to compare between species and F2 using Figure 3 and 4. I would suggest combining these figures and organizing image panels much like was done for Figure 2, where it is much easier to compare between the two species and the F2.

Third, it was a bit odd to me to first have the genetic map and QTL plot shown, followed by all of the traits described and measured. I feel like it would be more effective, and improve flow, to first describe and illustrate all of the phenotypes, followed by the map and corresponding results.

Finally, I think the discussion could use substantial reorganizing. The beginning of the discussion (lines 253-263) immediately lists limitations and caveats of this study system, and even the following section (lines 264-273) is written to support some of the caveats. While I appreciate the authors' transparency in discussing limitations, beginning the discussion weakens the interesting biology and impact of the study. I would suggest saving that discussion for closer to the end of the

discussion. Start with a stronger paragraph that outlines the major results. Lines 314-323 with some modification, might be much more suitable to lead with. Then get into the finer details of the results.

ADDITIONAL MINOR COMMENTS & QUESTIONS

Lines 39: What is meant by "rank"?

Lines 43-44: Change "differ from each microbiome" to "differ in their microbiome".

Lines 49-50: This is a very interesting fact, but the sentence needs to be rephrased because it currently reads awkwardly. Also, I'm not sure "phyllodia" has been defined at this point, so one should be added here.

Lines 65-66: Point (b) needs to be rephrased because it reads very awkwardly.

Lines 94-95: Do the authors have an explanation for why they found 42 linkage groups, when these species only have 13 chromosomes. I understand some small discrepancy, but 3-4x the number seems odd.

Figure 3 and 4: The authors will probably need to add actual trait names to either the figure, or to the figure legends.

Lines 254-255: I'm not sure the word "Mendelian" is necessary in this sentence.

Lines 324: The word "pileups" is not a common way to describe QTL that co-localize. I would suggest choosing a different term such as "co-localize" or "overlap".

Lines 326-327: I'm not sure I understand this sentence - why would it not be obvious that these traits are correlated? Many studies have demonstrated strong phenotypic and genotypic correlations between traits on the same general structure, such as, for example, leaf traits. I would have actually found it odd if the authors did not observe this (which you do, based on many QTL overlapping). The authors could easily calculate pairwise phenotypic (and/or genotypic) correlations for your 17 traits, and use this as evidence to support (or refute) trait correlations. This would be a nice supplemental addition to the QTL map that has several hot spots.

Lines 327-330: I would suggest defining the terms (pleiotropy, selective sweeps, chromosomal rearrangements) in this sentence, and explain how each can lead to overlapping QTL, as the readership may not be familiar with all of them. In addition, physical linkage should be added to that list.

Lines 336-337: Remove the word "but" at the end of this sentence.

Lines 354-370: Some of this interesting biology could be used to strengthen the introduction. Also, I'm surprised that there isn't a broad discussion about how the traits focused on in this study could affect things such as carnivory.

Lines 372-374: Again, this would be a good way to launch the introduction.

Lines 374-376: Here the authors mention "potential" in this system - why not provide an example of what is being referred to?

Lines 379-385: How would this benefit the community of biologists working on this, or a similar, study system?

Lines 389: Eggs?

Lines 453-454: The authors never discussed effect plots before this point, or how they were used to discern dominance. It might be beneficial to show at least a few exemplary effect plots in the supplemental materials.

Reviewer #2 (Comments to the Authors (Required)):

This manuscript describes a novel (and first) perennial F2 population of *Sarracenia* derived from a cross between 2 morphologically-diverged species, and its validation by mapping QTLs for pitcher traits.

The genetic map is not finished (as can be expected in the absence of decent parental reference sequences) but sufficient to start decomposing traits' variation into individual loci. It is certainly difficult to phenotypically quantify such complex and contrasted structures as pitchers. At the minimum, finding highly significant QTLs is a proof that some important variation has been captured at least indirectly, and that these traits define the different pitcher strategies. Other unshown traits failed to report anything significant. Overall, as always with 'simple' mapping publications, and although they represent a lot of work, no major new biological insights is gained, but this is an interesting resource and dataset for the future.

My main comment (detailed below) is that authors need to report on QTL allelic effect directions (not only dominance) as this is a major information to help make sense of the QTLs detected.

Specific points:

- L.16: "we found the genetic basis for..."

I would refrain from using this wording which may let readers believe that genes were identified behind the QTLs. This is not the case (and not requested at this stage, obviously). I would say "we mapped the genetic basis for...".

- Fig.1 legend: "additionally groups 31, 34, 40 are shown"

Figure 1 shows additional group 39 but not 40.

- Fig.3&4: I don't find these figures in the current form very useful (especially Fig.4) as the trait name indicated on them is rarely understandable or meaningful, and doesn't provide a real comparison.

What is for instance the point of figure 4F ? 4G is described only late in the discussion.

Rather show typical images of contrasted lines for some examples of well explained traits ?

- A table explaining the trait name code would be desirable (not just in the text or legends).

- Tab.1&2 & Results:

It might be a matter of taste, but I find it odd to present the results starting with the less significant

ones (in other words, the more likely to be false positives)...

- Tab.1:

I am surprised to find that the QTL for radalsym on LG1, with a LOD Score of 1.84, can be considered as significant. Usually, permutations for LOD Score tests rarely give genome-wide thresholds below 2.3LOD. I expect too high type-I error rate here. Please use permutation to define LOD thresholds.

There is one crucial information missing (usually reported together with the PVE): it is the direction of the allelic effect, in other words whether the rosea allele is increasing (or decreasing) the trait value with respect to the psittacina allele (this is of course different from the dominance effect reported in Tab.1). In addition to dominance, this is essential to -for instance- compare individual QTLs effect and parental phenotypic difference, or help to make sense of QTLs colocalization.

- Results & Discussion:

Please discuss QTLs results in the context of the direction of the allelic effect. For instance for any trait where *S. rosea* and *S. psittacina* are extreme in terms of phenotype, do all QTLs match (= contribute to) the expected direction of effect, or are some individual QTLs in the 'unexpected' direction? This information (which is anyway available to the authors from the QTL analysis) is strangely lacking from the results' discussion.

I could only find it used once when discussing a specific case of epistasis (L.167).

- Tab.2:

Similarly to Tab.1, has the significance threshold been adjusted for multiple testing? I find that some of the reported epistasis seem only very marginally significant.

- L.337: correct ", but."

- L.452: correct "significance"

A Carnivorous Plant Genetic Map: Pitcher/Insect-Capture QTL On a Genetic Linkage Map of *Sarracenia*
Russell L. Malmberg, Willie L. Rogers, Magdy S. Alabady

Responses to Reviewers' Requests and Comments

R> Overview of some changes (more details given below)

- Estimates of QTL effects have been added to what is now Table 2, and discussed as appropriate in the text. Thank you for pointing out this omission.
- The figures of the traits have been redone, creating a comparison for each trait. This led to generation of quite a few images. Following a suggestion of the Editor, these have been organized into a series of Figure Plates (numbers 4 to 9) each of which is about 1 page in size. The traits are pictured in decreasing order of %PVE similarly to the organization of the text and the new Table 1.
- There has been more information about *Sarracenia* added to the Introduction and Discussion.

Reviewer #1 (Comments to the Authors (Required)):

Major

1. I would suggest more strongly introducing the interesting ecology behind carnivorous adaptations in plants early in the introduction, as a means to attract your readership. The authors should discuss how these plants can thrive in low N soils, and expand on the phenotypic plasticity observed under varying levels of N (this is cool science, and would certainly add more interest). Likewise, they could expand on the tissue specialization brought up in lines 32-36.

R> The Introduction has had additional information on these topics added to it. We have also added material on why *S. rosea* and *S. psittacina* were chosen, and on some of the pitcher traits that differ between these species – as requested by the Reviewer in other points.

2. Furthermore, consider adding a few sentences describing why the two focal species were specifically chosen: Is there some evolutionary divergence story that may be of interest? Are these representative "model" species for carnivorous plants? In other words, what makes these two species of particular interest.

R> This material has been moved from the Results to the Introduction and expanded including descriptions of pitcher traits and phylogeny.

3. The next major item I wanted to touch on was overall organization and structure across the manuscript. First, I do not think that Table 1 is necessary for the main text. The authors present your QTL, their locations and intervals in Figure 1. Furthermore, they present the %PVE and interactions in the results text that describes each mapped trait. Table 1 seems mostly redundant, and can be tucked into the supplemental material.

R> Our preference is to include the table in the main text. The locations and intervals in Figure 1 are visually useful but not a substitute for the actual numbers; the redundancy with the text is needed to link the table lines and Figure to the text where the traits and their measurement methods are

described in detail. Virtually all QTL papers have an equivalent data table as their central focus, convenient for the casual or detailed reader and giving the primary data and statistics used. (The former table 1 is now table 2).

4. Secondly, it is difficult to compare between species and F2 using Figure 3 and 4. I would suggest combining these figures and organizing image panels much like was done for Figure 2, where it is much easier to compare between the two species and the F2.

R> These figures have been reworked in line with the suggestion; they are now Figures 4 to 9.

5. Third, it was a bit odd to me to first have the genetic map and QTL plot shown, followed by all of the traits described and measured. I feel like it would be more effective, and improve flow, to first describe and illustrate all of the phenotypes, followed by the map and corresponding results.

R> Information about pitcher traits has been placed in the expanded Introduction; this made the most sense to us. Within the Results and Discussion, sections we prefer the original results order of Map then phenotypes and QTL, rather than having phenotypes then map then more phenotypes.

6. Finally, I think the discussion could use substantial reorganizing. The beginning of the discussion (lines 253-263) immediately lists limitations and caveats of this study system, and even the following section (lines 264-273) is written to support some of the caveats. While I appreciate the authors' transparency in discussing limitations, beginning the discussion weakens the interesting biology and impact of the study. I would suggest saving that discussion for closer to the end of the discussion. Start with a stronger paragraph that outlines the major results. Lines 314-323 with some modification, might be much more suitable to lead with. Then get into the finer details of the results.

R> Three sentences were added at the beginning of the discussion to provide a better start to this section. The remaining portions were rewritten to reflect this change.

Minor

7. Lines 39: What is meant by "rank"?

R> This is a standard use of rank to indicate that whether a taxa is formally a species, subspecies, etc. is undetermined, but it is recognized as being a distinctive biological grouping in some sense. The use of the term species in reference to *Sarracenia* is problematic due to frequent hybridization; there is not a good species concept for the genus nor a satisfactory delineation of species. The original text is unchanged.

8. Lines 43-44: Change "differ from each microbiome" to "differ in their microbiome".

R> Done.

9. Lines 49-50: This is a very interesting fact, but the sentence needs to be rephrased because it currently reads awkwardly. Also, I'm not sure "phyllodia" has been defined at this point, so one should be added here.

R> Done.

10. Lines 65-66: Point (b) needs to be rephrased because it reads very awkwardly.

R> Done.

11. Lines 94-95: Do the authors have an explanation for why they found 42 linkage groups, when these species only have 13 chromosomes. I understand some small discrepancy, but 3-4x the number seems odd.

R> This is not unusual for a partially finished genetic map. The explanations are given in the text in the Introduction and Discussion – the large genome, the lack of a high-quality reference genome sequence, the gene duplications, starting with heterozygous parents. In many cases, potentially useful SNPs get discarded because orthologs and paralogs can't be distinguished. Plant genomes of this size are usually made up of a lot of retrotransposons and repetitive sequences.

We have done enough RARseq and RADseq sequencing and analysis runs to know that additional marker discovery sequencing will not yield additional informative markers without a high-quality reference genome sequence.

12. Figure 3 and 4: The authors will probably need to add actual trait names to either the figure, or to the figure legends.

R> Fixed in the revised figures 4 to 9.

13. Lines 254-255: I'm not sure the word "Mendelian" is necessary in this sentence.

R> Its not necessary to those with a classic genetic background, but there has been occasional recent use of "genetic map" in the scientific and popular literature to indicate that an organism has had its genome sequenced, so it is useful to make the distinction of type of genetic map. We have changed this to "genetic linkage map" . (*Upon reflection, the mention of Mendel might have been less appropriate than Sturtevant, so we just added "linkage" as the simplest solution*).

14. Lines 324: The word "pileups" is not a common way to describe QTL that co-localize. I would suggest choosing a different term such as "co-localize" or "overlap".

Done.

15. Lines 326-327: I'm not sure I understand this sentence - why would it not be obvious that these traits are correlated? Many studies have demonstrated strong phenotypic and genotypic correlations between traits on the same general structure, such as, for example, leaf traits. I would have actually found it odd if the authors did not observe this (which you do, based on many QTL overlapping). The authors could easily calculate pairwise phenotypic (and/or genotypic) correlations for your 17 traits, and use this as evidence to support (or refute) trait correlations. This would be a nice supplemental addition to the QTL map that has several hot spots.

R> The sentence has been deleted as poorly written. Traits and genes got confused in the original sentence, so its been deleted.

16. Lines 327-330: I would suggest defining the terms (pleiotropy, selective sweeps, chromosomal rearrangements) in this sentence, and explain how each can lead to overlapping QTL, as the readership may not be familiar with all of them. In addition, physical linkage should be added to that list.

R> A discussion of these terms has been added.

17. Lines 336-337: Remove the word "but" at the end of this sentence.

R> Oops. Done.

18. Lines 354-370: Some of this interesting biology could be used to strengthen the introduction. Also, I'm surprised that there isn't a broad discussion about how the traits focused on in this study could affect things such as carnivory. Lines 372-374: Again, this would be a good way to launch the introduction.

R> These sentences were moved to the Introduction

19. Lines 374-376: Here the authors mention "potential" in this system - why not provide an example of what is being referred to? Lines 379-385: How would this benefit the community of biologists working on this, or a similar, study system? Lines 389: Eggs?

R> Rewritten with some "potential" possibilities included

20. Lines 453-454: The authors never discussed effect plots before this point, or how they were used to discern dominance. It might be beneficial to show at least a few exemplary effect plots in the supplemental materials.

R> Done. Supplemental Figure 1 contains example effect plots and this is mentioned in the text.

Reviewer #2 (Comments to the Authors (Required)):

Main

1. My main comment (detailed below) is that authors need to report on QTL allelic effect directions (not only dominance) as this is a major information to help make sense of the QTLs detected.

R> Done. Thank you for pointing this out.

Specific

1. - L.16: "we found the genetic basis for..." I would refrain from using this wording which may let readers believe that genes were identified behind the QTLs. This is not the case (and not requested at this stage, obviously). I would say "we mapped the genetic basis for..."

R> Done. phrase changed to "mapped the genetic basis"

2. - Fig.1 legend: "additionally groups 31, 34, 40 are shown" Figure 1 shows additional group 39 but not 40.

R>. Oops. Done

3. - Fig.3&4: I don't find these figures in the current form very useful (especially Fig.4) as the trait name indicated on them is rarely understandable or meaningful, and doesn't provide a real comparison. What is for instance the point of figure 4F ? 4G is described only late in the discussion. Rather show typical images of contrasted lines for some examples of well explained traits ?

R> The figures have been reworked.

4. - A table explaining the trait name code would be desirable (not just in the text or legends).

R> Done. Good idea. This is now table 1.

5. - Tab.1&2 & Results: It might be a matter of taste, but I find it odd to present the results starting with the less significant ones (in other words, the more likely to be false positives)...

R> Done. The text and tables have been re-sorted in decreasing order of %PVE

6. - Tab.1: I am surprised to find that the QTL for radalsym on LG1, with a LOD Score of 1.84, can be considered as significant. Usually, permutations for LOD Score tests rarely give genome-wide thresholds below 2.3LOD. I expect too high type-I error rate here. Please use permutation to define LOD thresholds.

R> Permutations, 2000, were used in the analysis; we should have reported this in the M&M. This has been fixed. Radalsym is the trait for which the data is weakest, but its one locus still fell under a predetermined threshold probability of significance; hence it is appropriate to report it. Its an interesting phenotype, apparent in the F2 and not the parents, and we hope that a future researcher

will be inspired to find growth conditions which improve its heritability and better define its genetic basis.

7. There is one crucial information missing (usually reported together with the PVE): it is the direction of the allelic effect, in other words whether the rosea allele is increasing (or decreasing) the trait value with respect to the psittacina allele (this is of course different from the dominance effect reported in Tab.1). In addition to dominance, this is essential to -for instance- compare individual QTLs effect and parental phenotypic difference, or help to make sense of QTLs colocalization.

Please discuss QTLs results in the context of the direction of the allelic effect. For instance for any trait where *S. rosea* and *S. psittacina* are extreme in terms of phenotype, do all QTLs match (= contribute to) the expected direction of effect, or are some individual QTLs in the 'unexpected' direction ? This information (which is anyway available to the authors from the QTL analysis) is strangely lacking from the results' discussion. I could only find it used once when discussing a specific case of epistasis (L.167).

R> Done. Thank you for pointing this out.

8. Similarly to Tab.1, has the significance threshold been adjusted for multiple testing ? I find that some of the reported epistasis seem only very marginally significant.

R> The reviewer is correct that we did not adequately correct for multiple hypothesis testing for the epistasis tests. This has been fixed both in the table and text. The primary change is the heigwing potential interactions are no longer reported. We thank the reviewer for pointing this out.

9. - L.337: correct ", but."

R> Oops. Done.

10. - L.452: correct "signficance"

R> Oops. Done.

November 13, 2018

RE: Life Science Alliance Manuscript #LSA-2018-00146-TR

Dr. Russell L Malmberg
University of Georgia
Plant Biology Department
2502 Miller Plant Sciences Bldg
Athens, Georgia 30602-7271

Dear Dr. Malmberg,

Thank you for submitting your revised manuscript entitled "A Carnivorous Plant Genetic Map: Pitcher/Insect-Capture QTL On a Genetic Linkage Map of Sarracenia".

As you will see, the reviewers appreciate the revision performed, but think that the manuscript should be further improved prior to publication. We would therefore like to invite you follow the suggestions / requests for clarification of the reviewers and to provide a final version of your manuscript alongside a point-by-point response.

Please also make sure that you cite only articles that have been published or that are accepted for publication at a named publication. Wikipedia entries are subject to change, so should not be referred to. Please make sure to follow reviewer #1's suggestions on writing style (no bullet points, ease of read of introduction). Finally, please display in Figures 4-9 for each panel the trait and species, but move the further description that can be currently found underneath the images into the legends for each figure. The legends of each figure should call out all panels (eg as you do for Figure 3).

A. FINAL FILES:

-- High-resolution figure, supplementary figure and video files uploaded as individual files: See our detailed guidelines for preparing your production-ready images, <http://life-science-alliance.org/authorguide>

-- Summary blurb (enter in submission system): A short text summarizing in a single sentence the study (max. 200 characters including spaces). This text is used in conjunction with the titles of

papers, hence should be informative and complementary to the title. It should describe the context and significance of the findings for a general readership; it should be written in the present tense and refer to the work in the third person. Author names should not be mentioned.

B. MANUSCRIPT ORGANIZATION AND FORMATTING:

Full guidelines are available on our Instructions for Authors page, <http://life-science-alliance.org/authorguide>

Thank you for your attention to these final processing requirements.

Sincerely,

Reviewer #1 (Comments to the Authors (Required)):

I enjoyed re-reviewing this paper. I wanted to thank the authors for addressing all of the suggestions I provided, and/or providing a detailed response. I find this to be an interesting study, and of value to a growing community of scientists interested in studying non-model systems using genetics/genomics. I only have a few comments below.

The authors may want to check with the editors if the use of bullet points is acceptable for a publication in this journal, as was done in the introduction and discussion sections.

The authors may also want to check with editors if the use of Wikipedia is an acceptable source to cite (cited twice in the discussion).

I still believe that having trait QTL information presented in all of a figure (figure 1), a table (table 2), and described in detail in the text (results section) seems redundant to me. However, as the authors have argued, they feel that it is necessary to include all three in the main text. I guess this is just a matter of preference.

The introduction, while greatly improved by moving interesting background into it, is still a bit "clunky", and does not seem to flow very well. I apologize for being critical about this, but think that a bit more work on the introduction will improve the manuscript for readers. I would consider removing the use of bullet points, and trying sub-sections with descriptive headers that guide the readers.

Other than that I find the manuscript to be in good shape, and thank the authors for their work.

Reviewer #2 (Comments to the Authors (Required)):

I am overall satisfied with the actions taken by the authors to consider my remarks, although some of the changes seem really minimal especially the discussion of allelic effect direction (but it is difficult to tell in the absence of the submission of a proper tracked-changes version of the text).

Just 2 additional remarks that the authors should consider when preparing their final version:

- I could not find an essential information with the reporting of allelic effects: How (= in which direction) were they estimated? In other words, are the reported values calculated as [S. psittacina - S. rosea] or the other way around [S. rosea - S. psittacina]? According to the results, it seems to be [S. rosea - S. psittacina] for some traits (periotuk, ptnwindw), but [S. psittacina - S. rosea] for others (openess, openfrac)! If this is not true, then there is a discrepancy (to be solved) between Table 2 and text (Results). In any case, it is not clearly mentioned (apologies if I just missed it), while this is crucial to make sense of the sign of the allelic effect.

- the new sorting of Table 2 makes me realize that the reported LOD values are very high compared to what is usually estimated with R/qtl scanone() functions. I would just suggest to the authors to double check for themselves that they did not accidentally report LRT values (usually about 5 times higher by construction) instead of LOD.

A Carnivorous Plant Genetic Map: Pitcher/Insect-Capture QTL On a Genetic Linkage Map of *Sarracenia*
Russell L. Malmberg, Willie L. Rogers, Magdy S. Alabady

Responses to Reviewers' Comments

R> We have added a thanks for the reviewers' suggestions to the acknowledgements.

Reviewer #1:

I enjoyed re-reviewing this paper. I wanted to thank the authors for addressing all of the suggestions I provided, and/or providing a detailed response. I find this to be an interesting study, and of value to a growing community of scientists interested in studying non-model systems using genetics/genomics. I only have a few comments below.

1. The authors may want to check with the editors if the use of bullet points is acceptable for a publication in this journal, as was done in the introduction and discussion sections.

R> These paragraphs were re-written to remove bullet points.

2. The authors may also want to check with editors if the use of Wikipedia is an acceptable source to cite (cited twice in the discussion).

R> Wikipedia citations were replaced with journal article citations.

I still believe that having trait QTL information presented in all of a figure (figure 1), a table (table 2), and described in detail in the text (results section) seems redundant to me. However, as the authors have argued, they feel that it is necessary to include all three in the main text. I guess this is just a matter of preference.

3. The introduction, while greatly improved by moving interesting background into it, is still a bit "clunky", and does not seem to flow very well. I apologize for being critical about this, but think that a bit more work on the introduction will improve the manuscript for readers. I would consider removing the use of bullet points, and trying sub-sections with descriptive headers that guide the readers.

R> This section was modified to improve the flow while removing the bullet points.

Other than that I find the manuscript to be in good shape, and thank the authors for their work.

Reviewer #2

I am overall satisfied with the actions taken by the authors to consider my remarks, although some of the changes seem really minimal especially the discussion of allelic effect direction (but it is difficult to tell in the absence of the submission of a proper tracked-changes version of the text).

Just 2 additional remarks that the authors should consider when preparing their final version:

4. I could not find an essential information with the reporting of allelic effects: How (= in which direction) where they estimated? In other words, are the reported values calculated as [S. psittacina - S. rosea] or the other way around [S. rosea - S. psittacina]? According to the results, it seems to be [S. rosea - S. psittacina] for some traits (periotuk, ptnwindw), but [S. psittacina - S. rosea] for others (openness, openfrac)! If this is not true, then there is a discrepancy (to be solved) between Table 2 and text (Results). In any case, it is not clearly mentioned (apologies if I just missed it), while this is crucial to make sense of the sign of the allelic effect.

R> Its $0.5 * (\text{Srosea-value} - \text{Spsittacina-value})$ as calculated by R/qtl. This has been added to the Materials and Methods. Thank you for the suggestion.

The figure below shows the R/qtl effect plots for 2 of the traits and loci questioned.

The effect value for openness-1 in table 2 is **-0.56**; this is consistent with the effectplot shown on the left, where the value of S. rosea (BB) is visually 1.3 and the value of S. psittacina (AA) is visually 2.4, with a difference of -1.1 and half that of -0.55

The effect value for periotuk-2 in table 2 is **+0.34**; the effect plot on the right shows S. rosea higher than S. psittacina and a visual estimate of a difference of close to 0.7

openness-1 and periotuk-2 Effect Plots. AA is S. psittacina and BB is S. rosea

5. The new sorting of Table 2 makes me realize that the reported LOD values are very high compared to what is usually estimated with R/qtl scanone() functions. I would just suggest to the authors to double check for themselves that they did not accidentally report LRT values (usually about 5 times higher by construction) instead of LOD.

R> We checked and the LOD scores shown are the LOD scores we obtained by R/qtl, not accidentally LRT. Yes, the value for numpitch in particular is high, but it is what was reported by the program.

RE: Life Science Alliance Manuscript #LSA-2018-00146-TRR

Dr. Russell L Malmberg
University of Georgia
Plant Biology Department
2502 Miller Plant Sciences Bldg
Athens, Georgia 30602-7271

Dear Dr. Malmberg,

Thank you for submitting your Resource entitled "A Carnivorous Plant Genetic Map: Pitcher/Insect-Capture QTL On a Genetic Linkage Map of Sarracenia". It is a pleasure to let you know that your manuscript is now accepted for publication in Life Science Alliance. Congratulations on this interesting work.

DISTRIBUTION OF MATERIALS:

Again, congratulations on a very nice paper. I hope you found the review process to be constructive and are pleased with how the manuscript was handled editorially. We look forward to future exciting submissions from your lab.

Sincerely,
